



# Glacier Changes in the Chhombo Chhu Watershed of Tista basin between 1975 and 2018, Sikkim Himalaya, India

Arindam Chowdhury[1], Milap Chand Sharma[2], Sunil Kumar De[1], and Manasi Debnath[1]

[1]Department of Geography, North-Eastern Hill University, Shillong - 793022, Meghalaya, India

[2]Centre for the Study of Regional Development, Jawaharlal Nehru University, New Delhi 110067, India

*Correspondence to:* Sunil Kumar De (desunil@yahoo.com) Phone: (off): +91 364 2723205; (mob): +91 9862009202. Fax: +91 364 255 0076.

**Abstract.** Glaciers of the Tista basin represent an important water source for mountain communities and large population
downstream. The present article attempts to assess the observable changes in the glacier area in the Chhombo Chhu Watershed (CCW) of Tista basin, Sikkim Himalaya. The CCW consists of 74 glaciers (>0.02 km$^2$) with a mean glacier size of 0.61 km$^2$. The change of such glacier outlines obtained from the declassified hexagon KH-9 (1975), Landsat 5 TM (1989), Landsat 7 ETM+ (2000), Landsat 5 TM (2010), and Sentinel 2A (2018). The total glacier area in 1975 was 62.6 ± 0.7 km$^2$; by 2018, the area had decreased to 44.8 ±1.5 km$^2$, an area loss of 17.9 ± 1.7 km$^2$ (0.42 ± 0.04 km$^2$ a$^{-1}$). Debris free
glaciers exhibit more area loss by 11.8 ± 1.2 km$^2$ (0.27 ± 0.03 km$^2$ a$^{-1}$) followed by partially debris-covered (5.0 ± 0.4 km$^2$ or 0.12 ± 0.01 km$^2$ a$^{-1}$) and maximum debris-covered (1.0 ± 0.1 km$^2$ or –0.02 ± 0.002 km$^2$ a$^{-1}$) glaciers. The quantum of glacier area loss in the CCW of Sikkim Himalaya took its pace during 2000–2010 (0.62 ± 0.5 km$^2$ a$^{-1}$) and 2010–2018 (0.77 ± 0.6 km$^2$ a$^{-1}$) timeframes. Field investigations of selected glaciers and climatic records also support the trend in glacier recession in the CCW due to a significant increase in temperature trend and more or less static precipitation since
1995. Glacier retreat rates in the CCW were almost similar to the Changme Khangpu basin and other selected glaciers in Sikkim Himalaya. This glacier inventory and area change analysis will provide valuable information to the glaciological and hydrological community to model and plan the water resources in the Sikkim state of Eastern Himalaya. The dataset is now available from the Zenodo web portal: http://doi.org/10.5281/zenodo.4457183 (Chowdhury et al., 2021).

**Keywords.** Glacier changes; debris-free glaciers; glacier classification; elevation effects; climate change; Sikkim

## 1. Introduction

The best manifestation of the climate change, either positive or negative, is the advance and retreat of mountain glaciers in the Hindu-Kush-Himalayan (HKH) and Tibetan region that is often called as 'Third Pole' region of the world. Both these processes are time dependent, for it will be determined by the size of a glacier and the related influencing factors.
Glaciers play an important role as response-indicators of climate change (Bahuguna et al., 2014).

Most of the Himalayan glaciers are losing mass at different rates from each other as well as from other parts of the globe. Exceptional trends in glaciers in the central and western part of Karakoram Himalaya also prevails (Bajracharya and Shrestha, 2011; Hewitt, 2011; Mehta, 2011; Benn et al., 2012; Bolch et al., 2012; Bhambri et al., 2013; Bahuguna et al., 2014; Bajracharya et al., 2014; Qureshi et al., 2017; Raj et al., 2017). However, the future stability of such a trend on
glacier status is debatable under different climate scenarios (Ruosteenoja et al., 2003; Immerzeel, 2008). Since 1901, a significant rise in daily maximum temperature of 1°C in North-East India and basin-wide warming trend of 0.6°C in the Brahmaputra basin have been documented (Dash et al., 2007; Immerzeel, 2008). Moreover, the station-based

meteorological data from north-eastern part of India and Sikkim indicates that the changes in rainfall and temperature pattern are seasonal as well as site specific (Sreekesh and Debnath, 2016; Kumar et al., 2020a). Future climate projection

scenario reported substantial rise in temperature and imbalances in precipitation regimes that may aggravate the risk related to future glacier melt run off security amidst the HKH and peripheral Asian countries (Kumar et al., 2006).

However, the ongoing changes in the glaciers and snow cover due to global climate change are markedly affecting hydrological regimes in high-elevation mountain catchments in the HKH as well as worldwide. Most importantly, the run-off from these glaciers of the HKH region supports a vast population of about 1.3 billion people in its downstream basins

of Asian countries (Williams, 2013). Therefore, changes in glacier-melt runoff will, directly and indirectly, affect roughly a fifth of the world's population. Cryosphere dynamics affects not only the water cycle but also the sediment supply to the channel network. Thus, the sensitivity of glacier-fed mountain channels to climate change has a high-risk factor on the hydroelectric power stations, sediment supply over the flood plain, etc. Upper Tista basin that constituted the Sikkim Himalaya has vast potential for hydropower development and has a 5353 MW installed capacity that is 60% of the basin's

total hydropower potential (Khawas, 2016; Rahaman and Mamun, 2020). Many hydropower projects are already under construction, and some are planned to be established in North Sikkim (EDPS, 2019).

At the local level basin study, glacier variability assessment can help in analysing the glacier melt runoff-sediment budget that sustains the numerous peripheries of the population across the international boundary (Kumar et al., 2020b). Being a trans-boundary river, the glacier melt runoff from the upper reaches of River Tista in India also sustain the

population at its lower reaches in Bangladesh (Khawas, 2016; Rudra, 2018; Rahaman and Mamun, 2020). Therefore, historical change analysis and monitoring of glaciers in the Upper Tista basin are of utmost necessity for planning water resources from the national to international level.

Such local-level glacier monitoring attracts more attention because climatic parameters and temperature changes synchronize with local topographic parameters (Pepin et al., 2015). Even orographic variation determines the amount of

precipitation in the form of rainfall and snowfall in the Himalayan region (Singh and Kumar, 1997). Similarly, glacier distribution and anomaly in glacier changes also depend on the local topography and local climate over the HKH region (Kääb et al., 2007; Hewitt, 2011; Brahmbhatt et al., 2017; Ojha et al., 2017). Glacier responses, additionally, varied as per glacier's surface topography (debris-covered or clean ice ablation zone), presence of supraglacial and proglacial ponds, surface temperature, the morphology of glaciers, surrounding potential debris supply zone, etc. (Sakai et al., 2000; Scherler

et al., 2011; Ojha et al., 2017; Salerno et al., 2017; Olson and Rupper, 2019; Tsutaki et al., 2019). These widely varied parameters forced us to conduct catchment-based monitoring of glaciers in Sikkim Himalaya for the last couple of decades.

Unfortunately, the monitoring of glacier variability has generally been lacking in eastern Himalaya, and little attention has been paid to assess the catchment-based glacier status. Few studies have been conducted on Bhutan Himalaya, the entire Sikkim Himalaya, and the Changme Khangpu basin in Sikkim Himalaya. The Chhombo Chhu watershed (CCW)

containing a large number of glaciers of Sikkim is still unraveled. Hence this study has focused on three primary objectives: (i) Preparation of a detailed glacier inventory for CCW in Sikkim Himalaya using Sentinel 2A MSI (2018), (ii) Analyze the glacier area changes since 1975, (iii) Impact of topographic and non-topographic (*climatic factors*) parameters on glaciers since historical past.

## 2. Study Area

The Chhombo Chhu watershed (*"Chhu"* literally means water, but streams in the Himalayan Buddhist language *are*

*termed as "Chhu", and* a lake*, "Tso or Chho"*) is located (N 27°45'19.2" to N 28°7'41.2" and E 88°26'49.9" to E 88°50'45.9") in the upper Tista river basin in Sikkim in the Eastern Himalayas (Fig. 1). It covers a total surface area of 694.5 km$^2$, ranging in altitudes from ~2680 m to 7073 m above sea level. The Chhombo Chhu originates from Khangchung Tso, a proglacial lake fed by Tista Khangse glacier in the extreme northeast part of this watershed. Many other glaciers

feed lakes, such as Tso Lhamu and Gurudongmar supply meltwater throughout the year to the Chhombo Chhu which drains through the Thangu valley to join Lachen Chhu at Zema (2680 m.a.s.l). Finally the Lachen and Lachung Chhu confluence at Chungthang (1535 m.a.s.l), thereafter known as River Tista, an important tributary of the Brahmaputra in the Eastern Himalayas (CISMHE, 2007; Basnett et al., 2013; Debnath et al., 2018). Some of the important glaciers in this watershed are Tista Khangse (TK), Gurudongmar, Kangchengyao, Chhuma Khangse, Tasha Khangse and Yulhe Khangse.

85                                         Figure 1

It is now known that the Indian Summer Monsoon (ISM) plays a pivotal role in the glacier fluctuations in the entire Sikkim Himalaya (Racoviteanu et al., 2015) and additionally receive occasional precipitation through North-east (winter) monsoon as well as mid-latitude westerlies (Ali et al., 2018). At the higher elevations most of the precipitation, at any time of the year, is in the form of snow. The study region (CCW) is governed by two types of climatic conditions, i.e., the higher

reaches of the mountain to the north of Thangu valley and the northern slope of Kangchengyao Massif is dominated by the cold semi-arid deserted climate; whereas the lower part of the watershed from Thangu is characterized by Temperate climate (Debnath et al., 2018). The higher reaches of the mountain system to the north is an extension of Trans-Himalaya of Tibet region, and is comparatively cold semi-arid deserted type climate, is similar to Ladakh region in the west (Brazel and Marcus, 1991), and it is nearly devoid of vegetation, except for the little shrubs close to Gurudongmar and Tso Lhamo

lakes. Glaciers on the northern slope of Kangchengyao massif are in a rain-shadow area, having scanty precipitation, whereas the glaciers on the southern slopes of the massif receive abundant precipitation from the ISM during June to September. Therefore, it is assumed that the glaciers of the Sikkim Himalaya are generally summer accumulation irrespective to the western Himalaya and Karakoram region (Ageta and Higuchi, 1984; Debnath et al., 2019). The climograph (Fig. 2b) shows a long term average temperatures that does not exceed 10°C, whereas, mean monthly

precipitation ranges from 1.3 mm to 104 mm based on the nearest meteorological station data of Pagri (54 km from the centroid of the basin) (Fig. 2a).

Figure 2

Figure 3

### 3.  Materials and Methods

**3.1.  Data sources**

A variety of remotely sensed satellite images, with different temporal, multi-spectral and medium to high spatial resolution have been used to delineate glacier outlines, and for the change detection analysis in the study region (Table 1). For this study, these images have been selectively chosen only for the end of ablation period, with minimal seasonal snow and cloud coverage (November to December) and a vertical solar position to avoid shadows (Chand and Sharma, 2015).

These images include; panchromatic declassified hexagon (KH9-11; 1975), Landsat 5 Thematic Mapper (TM; 1988 and 2010), Landsat 7 Enhanced Thematic Mapper Plus (ETM+; 2000) and Sentinel 2A (2018), and further cross-verified with



the high resolution images on the Google Earth (GE) platform. In addition, the Survey of India (SoI) topographical sheets at 1:50,000 scale with 40 m contour interval have been used for obtaining topographic information; and Shuttle Radar Topography Mission (SRTM; 2000) Digital Elevation Model (DEM) as a reference DEM for the delineation of drainage basins and extraction of glacier topographic parameters, respectively (Table 1). For our study area, it is statistically proven that the SRTM (30 m) has an overall edge over ASTER and CARTOSAT-1 DEMs (Debnath et al., 2018). All the satellite images and SRTM DEM are freely obtained from the U.S. Geological Survey (USGS; http://earthexplorer.usgs.gov/). Whereas, the monthly average temperature (°C) and precipitation (mm) for Pagri Meteorological Station (PMS) from 1960–2013 was obtained from Climate Research Unit (CRU), University of East Anglia (http://www.cru.uea.ac.uk/data).

Table 1

### 3.2. Radiometric and geometric correction of satellite images

By the end of 1960s, when Corona reached its technical limits, a series of photographic reconnaissance satellites (KH9) with higher spatial resolution and larger coverage were launched by U.S. between 1971 and 1984 (Surazakov and Aizen, 2010). In this study, the 10 small subsets of declassified hexagon (KH9-11) image (original frame ~ about 250×125 km on the ground) acquired during KH9 mission (1211) in 1975, has been spatially georeferenced and co-registered to Sentinel 2A (S2A) (base image) using Spline Transformation Method (STM), along with adjustment rectification algorithm, to reconstruct the correct image geometry that existed at the moment of film exposure. All of the semi-automated geometric calibration procedures have been processed in ESRI ArcGIS 10.2.2. Software. We have acquired 286 GCPs all over the subset image that surrounds the glacier and snow-covered mountain range, which consists of prominent peaks, stable river junctions and roads, unchanged lineament structures, lake shores and rock outcrops were used for image orientation (Bhambri et al., 2011). The quality of the KH9-11 image was extremely good as well as nearly scratch and noise-free. Fig. 4a shows the distortion fields of KH9-11 those have been noticed in the corner areas near the mountain ridges similar to that of previous studies (Surazakov and Aizen, 2010) and later geometrically rectified Fig. 4b. The horizontal shift or positional accuracy between KH-9 and S2A images has been estimated at ~3.8m (~0.94 pixels).

Being a fundamental step in radiometric preprocessing, atmospheric corrections and topographically-induced illumination variations, all the different date datasets of Multi-spectral (MS) bands of Landsat series have been converted from DN values into a common meaningful physical unit such as spectral radiance ($L_\lambda$) and top of atmospheric (TOA) reflectance ($\rho_p$) (Chuvieco and Huete, 2009; Debnath et al., 2018). Finally, the dark object subtraction (DOS) method has been adopted for the radiometric correction of each image (Chavez, 1988) using ENVI 5.1. Software. All the stacked Landsat images have been co-registered with the orthorectified S2A image. The horizontal shift of Landsat TM and pan-sharpened ETM+ were calculated at ~11.7 (~0.39 pixels) and ~10.2 m (~0.68 pixels), respectively to the base image. Finally, all the visible, NIR and SWIR bands of S2A image has been resampled to 10 m resolution and layer stacked in ESA SNAP 5.0. Software to convert into a Geo-tiff image file for the extraction of updated glacier inventory (2018) in the CCW.

Figure 4





### 3.3. Glacier inventory mapping

Glacier boundaries have been manually delineated from the KH9-11 (1975) panchromatic scene (Fig. 4b). Algebraic
algorithms for image enhancement (viz. NDVI, NDWI and NDSI etc.) have been applied on different spectral bands of the
Landsat and Sentinel 2A images to semi-automatically map of clean glacier ice (Bolch and Kamp, 2006; Racoviteanu et
al., 2008, 2009; Bhambri et al., 2011). However, these algorithms couldn't differentiate the debris-covered glaciers
correctly from the surrounding moraines or rock outcrops in the region, therefore, have been manually digitized (Paul et
al., 2004; Racoviteanu et al., 2009; Bhambri et al., 2011; Frey et al., 2012). Additionally, high resolution Google Earth (≤5
m), Sentinel 2A (10 m), PAN-sharpened multispectral images of Landsat ETM+ (15 m), and the outcome of topographic
parameters such as slope, aspect and hill shaded relief maps, calculated from SRTM DEM, have been used for visual
rectification and check to map the glaciers (Schmidt and Nüsser, 2012). The seasonal snow cover and shadow areas have
also been eliminated. The determination of glacier termini and glacial lakes of some important large valley glaciers (e.g.
Tista Khangse, Gurudongmar and Kangchengayao-2) have been mapped during multiple field expeditions, and repeat
photographs taken between 2017 and 2018 (Fig. 3). Finally, the glacier vector outlines (≥ 0.02 km²) were prepared in our
inventory (Chand and Sharma, 2015; Debnath et al., 2019).

The impact of topographic control on glacier fluctuations and distribution have been statistically evaluated. We have
also modified the morphological classification of glaciers in the watershed using the Global Land Ice Measurements from
Space (GLIMS) guidelines and Glacier Atlas of India (Rau et al., 2005; Raina and Srivastava, 2008; Chand and Sharma,
2015). The classified morphological glaciers are categorized as Valley (Simple basin; SB), Valley (Compound basin; CB),
Mountain (Simple basin, SB), Cirque, Niche and Glacieret (Ice aprons; IA & Snowfields; SF). In this study, Rock glaciers
(RG) are excluded for the reason as suggested earlier (Debnath et al., 2019).

### 3.4. Glacier change and uncertainty estimation

Topographical inventory parameters (e.g. glacier size, mean and median elevation, mean slope, aspect) of the glacier
outlines for the year 1975, 1988, 2000, 2010 and 2018 have been computed from the SRTM DEM using ArcGIS 10.2.2.
Software (Paul et al., 2002; Paul and Svoboda, 2009). Furthermore, the characteristics of glacier distribution and
fluctuations have been examined with the help of different statistical derived diagrams, and by analyzing the relationships
between topographic parameters and glacier outlines. Glacier area dynamics have been computed for 1975 to 2018 and
divided into four different periods: 1975–1988, 1988–2000, 2000–2010, and 2010–2018. Glacier area changes have been
directly calculated by subtracting the total area of the recent year (2018) from the initial year (1975). Absolute and relative
changes have also been calculated for these periods. Fig. 12 is a visual representation of different glacier's area change in
the study region.

Mapping uncertainty estimation is necessary to assess the significance of the results to avoid misinterpretation of
mapping of the glacier area. For each glacier, the error has been calculated based on a buffer drawn around the outlines of
the glacier using ArcGIS 10.2.2. Software, as suggested by Granshaw and Fountain (2006) and Bolch et al. (2010a,b). For
example, 5 m of buffer size (i.e. ½ of a pixel) has been drawn around the original glacier outlines of Sentinel 2A image.
Similarly, in the case of Landsat 5 (TM), 15 m buffer size have been used for the glacier outlines. For pan-sharpened
Landsat 7 ETM+ and KH9-11 images, the buffer size were 7.5 m and 2 m, respectively. It is also observed that larger
glacier outlines had relatively very small errors than the small glacier patches (Bolch et al., 2010a). In this study, the
mapping uncertainty of the total glacier area were calculated as ±0.7 km² (~1.2 %), ±5.3 km² (~8.9%), ±2.6 km² (~4.5%),



±4.8 km$^2$ (~9.5%), and ±1.5 km$^2$ (~3.4%) mapping uncertainty for the images of KH9-11 Hexagon (1975), Landsat TM 5 (1988), pan-sharpened Landsat ETM+ (2000), Landsat TM 5 (2010), and Sentinel-2A (2018) respectively. Glacier area change uncertainty ($\varepsilon_{ac}$) was also estimated following Eq. (1) (Hall et al., 2003):

$$\varepsilon_{ac} = \sqrt{(e_1)^2 + (e_2)^2} \tag{1}$$

where, $e_1$ and $e_2$ are the estimated errors associated with the glacier area of two different time periods. In generally, most of the area changes are restricted to the termini of the large glaciers than the upper part of the glaciers in the study area.

### 3.5. Climate trend estimation

The long-term (1960–2013) CRU monthly mean temperature and precipitation data of the Pagri Meteorological Station (PMS) (4330 m), located approx 54 km east from the centroid of the basin have been incorporated for the climate
trend analysis (Fig. 2a). The Mann–Kendall (MK) statistical method has been employed to assess the climatic trend of mean monthly temperature and precipitation, and its impact on glacier fluctuation in the study region (Kendall, 1975; Mann, 1945). A positive value of Mann–Kendall Z statistic ($Z_{MK}$) indicates an increasing trend and vice versa (Cengiz et al., 2020). The magnitude of a trend (i.e. rate of change per year) for time series analysis has been carried out using Sen's slope ($Q$) method (Sen, 1968).

### 3.6. Field measurement

Extensive field measurements have been conducted during the pre-monsoon (May–June) and post-monsoon (October– November) seasons between 2017 and 2018 for the validation of maps of select glaciers (e.g. Tista Khangse, Gurudongmar, Kangchengyao-2 and many unknown cirques and niche glaciers) of the study region. It is observed that the northern slope of Kangchengyao massif have large valley glacier mostly without much debris-cover except Kangchengyao-2 glacier (Fig.
3a2), with enormous proglacial lakes, crevasses, and braided glacial streams. We did not measure the termini positions of the above-mentioned glaciers, as these are connected with large and dangerous proglacial lakes (see Fig. 1b for location). Photographs taken of recent termini positions (2017-18) are from the outlet location of the moraine-dammed lakes (Fig. 3). The benchmark glacial lakes and associated morphology have been verified using a handheld Garmin *e*Trex 30x GPS with positional accuracy of ± 3 m (WAAS-enabled).

## 4. Results

### 4.1. Glacier inventory and its distribution in 2018

In this study, we have identified and mapped 74 glaciers larger than 0.02 km$^2$ (minimum size threshold) using Sentinel 2A MSI (2018) and corresponding Google Earth platform images that cover an area of 44.8 ± 1.5 km$^2$ (Table 2). The current study is the latest glacier inventory map (2018) of CCW in Sikkim Himalayas (Fig. 1b). Glaciers in the CCW range
from small glacieret to large valley types, with a size range from 0.02 to 6.7 km$^2$. The histogram and normal Q-Q plot suggest that the glacier area in the CCW is not normally distribution (Fig. 5a-b). The higher values of skewness (3.96) and kurtosis (19.32) and Shapiro-Wilk normality test (0.53) at 0.00 significance level also confirms the previous assumption. Median size of glacier area is 0.31 km$^2$ in the watershed (Fig. 5c), similar to the other glacierized basins in the Sikkim Himalaya (Racoviteanu et al., 2015; Debnath et al., 2019).

The mean glacier size of the watershed is 0.61 km$^2$ that is almost similar to other glaciated basins of Himalaya; e.g.,



Ravi (0.6 km$^2$), Changme Khangpu (0.9 km$^2$), Ladakh (1 km$^2$) but comparatively much smaller than Chenab (1.15 km$^2$), Jankar (1.2 km$^2$), Zemu (1.24 km$^2$), Shyok (1.4 km$^2$), Baspa basin (1.7 km$^2$) Saraswati/ Alaknanda basin (3.7 km$^2$) (Bajracharya and Shrestha, 2011; Bhambri et al., 2011; Frey et al., 2012; Schmidt and Nüsser, 2012; Chand and Sharma, 2015; Mir et al., 2017; Das and Sharma, 2018; Debnath et al., 2019). Tista Khangse is the largest glacier with an area of

6.7 ± 0.1 km$^2$, which contributes as a prime source and origin of Tista River in the northeastern most corner of the watershed in Sikkim. The glacier area are divided into five different size classes such as <0.5, 0.5–1, 1–1.5, 1.5–2 and >2 km$^2$ (Table 2; Fig. 6a). Out of these, <0.5 km$^2$ glacier size class has the maximum number of glacier count (51) with an area of 10.55 ± 0.6 km$^2$ followed by 13, 4, 1, and 5 glaciers in the size class of 0.5–1, 1–1.5, 1.5–2 , and >2 km$^2$ covering an area of 8.84 ± 0.3, 4.86 ± 0.2, 1.86 ± 0.04, and 18.7 ± 0.4 km$^2$ respectively. The glacier size class >2 km$^2$ are mostly dominated by

valley (CB and SB) and mountain glaciers (SB). All other topographically controlled parameters according to glacier size classes are mentioned in Table 2.

Out of the 74 count, maximum number of glaciers (64) are debris free, covering an area of 28.4 ± 1.1km$^2$, with a mean size of 0.4 km$^2$, followed by 07 partially debris covered (PDC) and 03 maximum debris covered (MDC) glaciers, with an area of 14.8 ± 0.4 km$^2$ and 1.6 ± 0.1 km$^2$, respectively. The mean size of PDC and MDC are 2.1 km$^2$ and 0.5 km$^2$.

Debris types according to size classes are tabulated as in Table 2.

Figure 5

Table 2

Based on the morphological classifications, valley (SB) glaciers are the dominant types in the CCW, covering an area of 17.9 ± 0.5 km$^2$ (40% of the total glacierized area), followed by mountain (SB) glaciers (13.3 ±0.5 km$^2$), valley (CB) glaciers (6.4 ± 0.1 km$^2$), glacieret (SF & IA) (2.9 ± 0.1 km$^2$), cirque (2.5 ± 0.1 km$^2$) and niche (1.7 ± 0.1 km$^2$) respectively. The mean size of different morphological types are given in Table 3. The pattern of morphological classification of glaciers in Sikkim (Eastern Himalaya) is different than that in the central and western Himalayan region (Raina and Srivastava,

2008). Most of the valley (SB) glaciers are debris free, with an area of 13.5 km$^2$ (75%), only 16% and 9% are PDC (2.9 km$^2$) and MDC (1.6 km$^2$) glaciers, respectively. Mountain (SB) glaciers are debris free with an area of 7.8 km$^2$ (59%), but 41% are PDC (5.5 km$^2$). Small glaciers (viz. cirque, niche and glacieret) are completely debris free in the watershed. Distribution of glaciers according to morphological types are also shown in the Fig. 9.

Topographic parameters such as elevation and slope also play crucial roles in the variations of regional characteristics

of glacier distribution (Bhambri et al., 2011). The mean elevation of the glaciers ranges from 4846 to 6691 m, with an average of 5598 m (Fig. 6a). The mean elevation is higher than that of the other basins of the central and western Himalayas, such as Alaknanda (5380 m), Bhagirathi (5544 m), Ravi (4828 m), Yamuna (5083 m), Sutlej (5436 m), Chenab (5064 m), Chandrabhaga (5373 m), Indus (5404 m), and Ladakh range (5497 m) (Bhambri et al., 2011; Frey et al., 2012; Schmidt and Nüsser, 2012; Chand and Sharma, 2015; Das and Sharma, 2018), which certainly reflects the fact that this study region

located at an extreme northeast part of Sikkim Himalaya has a cold semi-arid climatic condition. The studied glaciers have almost symmetrical average median elevation (5601 m) to mean elevation, and ranges from 4846 to 6666 m, which is also a good proxy for the long-term equilibrium-line altitude (ELA) estimations based on topographic parameters (Braithwaite and Raper, 2009) (Fig. 7a). The glacier termini are located around an average minimum elevation of 5348 m with varying





ranges from 4688 to 6529 m. Hence, large valley glaciers (SB) extend to the lower elevations, while smaller glaciers have
higher termini. All other topographic parameters (minimum and maximum elevation, average mean and range elevation)
vary according to the size class and morphological types (Table 2 & 3). Fig. 6b shows the distribution of glacier according
to the elevation size classes and morphological types.

Glaciers in the watershed are distributed with mean slope of 26°, ranging from 12° to 45°. It is clearly evident that
the larger glaciers have gentle slopes than the smaller one, and morphologically, the Mountain (SB), Cirque, Niche and
Glacieret (IA & SF) glaciers have steeper slope than valley (CB and SB) glaciers (Table 2 & 3; Fig. 6 & 7). Similar average
slopes were observed in the other basins of Sikkim in Eastern Himalaya (Racoviteanu et al., 2015; Debnath et al., 2019) as
well as in western Himalayas (Frey et al., 2012).

The area distribution by aspect sector shows that the glaciers are predominantly oriented towards north (~34%) and
followed by south (~15%) (Fig. 8a). Based on glacier size classes, the majority aspect is northern, except 1–1.5 km², which
have southwest orientation (Table 2). The valley (SB), cirque and niche glaciers are mainly oriented towards north with
54%, 38% and 81%, respectively. While the valley (CB), mountain (SB) and glacieret (SF & IA) have northeast aspect
(~55%), south (~49%) and southwest (~46%), respectively.

Table 3

Figure 6

Figure 7

Figure 8

Figure 9

## 4.2. Glacier area change (1975–2018)

Spatio-temporal change analysis reveals that the total glacier areal coverage across the entire study region in 1975
was 62.6 ± 0.7 km² (mean area: 0.75 km²) and, that by 2018, this area had reduced to 44.8 ± 1.5 km² (mean area: 0.61 km²)
(Fig. 10a). There is a total areal reduction of 17.9 ± 1.7 km² (~28.5 ± 3.6%) over the 43-year of analysis period (Table 4).
The annual rate of glacier loss tend to vary over the different timeframes, with an initial shrinkage rate of 0.24 ± 0.4 km²
a⁻¹ between 1975 and 1988, which is little higher than 1988 to 2000 (0.20 ± 0.5 km² a⁻¹). Between 2000 to 2010, shrinking
rate increased to 0.62 ± 0.5 km² a⁻¹, and this rate remained higher between 2010 and 2018 (0.77 ± 0.6 km² a⁻¹). The overall
rate of glacier loss between 1975 and 2018 is 0.42 ± 0.04 km² a⁻¹ (~0.66 ± 0.1% a⁻¹) (Table 4). The number in glaciers also
reduced from 83 (1975) to 74 (2018). Similar trend is reported in the Changme Khangpu basin (−0.45 ± 0.001 km² a⁻¹) of
Sikkim in the Eastern Himalaya between 1975 and 2016 (Debnath et al., 2019). It is ironical that the annual rate of glacier
loss has been found to be at a declining trend in the Ravi and Chenab basins of north-western Himalaya during 2001–2013
(Chand and Sharma, 2015; Brahmbhatt et al., 2017). Fig. 10c showing the retreat map of the study area (1975–2018). A
scatter plot compares the relationship of area of individual glaciers recorded in 1975 and 2018 (Fig. 10b). This diagram
shows that the individual glacier surface area has reduced, as well as 10 glaciers disappeared altogether over a period of
43 years in this watershed. Only one glacier (i.e., Chhuma Khangse-1) show fragmentation into two during the period
between 2010 and 2018 (Fig. 12a-b). Gurudongmar Khangse, a valley (SB) glacier, which was influenced by a large supra-
glacial lake (SGL-3) observed in 1975, later developed into a potentially dangerous moraine-dammed proglacial lake



(Worni et al., 2013), retreated from $3.7 \pm 0.03$ km$^2$ to $1.9 \pm 0.04$ km$^2$ ($-1.1 \pm 0.06\%$ a$^{-1}$) over the period between 1975 and 2018 (Fig. 12e-f).

Figure 10

Table 4


In total 72 glaciers were DF in 1975, which decreased to 64 in 2018. The DF glacier area decreased from $40.3 \pm 0.5$ km$^2$ (1975) to $28.4 \pm 1.1$ km$^2$ (2018), with an area change of $-11.8 \pm 1.2$ km$^2$ ($-0.27 \pm 0.03$ km$^2$ a$^{-1}$). The PDC also decreased from $19.8 \pm 0.2$ km$^2$ (1975) to $14.8 \pm 0.4$ km$^2$ (2018), an area change of $-5.0 \pm 0.4$ km$^2$ ($-0.12 \pm 0.01$ km$^2$ a$^{-1}$). Similarly, the glacier area change of MDC was $-1.0 \pm 0.1$ ($-0.02 \pm 0.002$ km$^2$ a$^{-1}$) since 1975 (Fig. 11e). During 1975–

2018, glacier size class of $>2$ km$^2$ lost maximum area of $-7.7 \pm 0.4$ km$^2$ ($-0.18 \pm 0.01$ km$^2$ a$^{-1}$) mainly due to lower terminus elevation. This is followed by size class of 0.5–1 km$^2$ ($-5.6 \pm 0.4$ km$^2$ or $-0.13 \pm 0.01$ km$^2$ a$^{-1}$), 1–1.5 km$^2$ ($-3.4 \pm 0.2$ km$^2$ or $-0.08 \pm 0.004$ km$^2$ a$^{-1}$), and 1.5–2 km$^2$ ($-1.4 \pm 0.1$ km$^2$ or $-0.03 \pm 0.001$ km$^2$ a$^{-1}$). On the contrary, the glacier size class of $<0.5$ km$^2$ has gained its area of $+0.3 \pm 0.6$ km$^2$ ($+0.01 \pm 0.01$ km$^2$ a$^{-1}$) since 1975. But in terms of relative figures, the size class of 1.5–2 km$^2$ lost maximum area of $-42.3 \pm 2.5\%$ ($-0.98 \pm 0.1\%$ a$^{-1}$) than any other size classes which is

mentioned in the Table 5.

The glacier area changes according to morphological types are tabulated in Table 6. It is clearly evident that valley (SB) glaciers have lost maximum percentage area of $-8.4 \pm 0.6$ km$^2$ ($-0.19 \pm 0.01$ km$^2$ a$^{-1}$) in the watershed than the other glacier morphological types. The position of lower terminus elevation and comparatively lesser slope of large valley (SB) glacier than the other morphological types were the reasons for maximum glacier area loss (Table 6; Fig. 11b-d). For

example, some notable large valley (SB) and (CB) glaciers have significantly retreated since 1975 such as Lachen Khangse-1 (~100%), Chombku Khangse (~55.1%), Gurudongmar Khangse (~49.1%), Tasha Khangse (~35.3%), Chhuma Khangse-1 (~31.3%), Kangchengyao-2 (~8.4%), Tista Khangse (~8.3%) in the study region.

This study records the maximum percentage of glacier area recession and its annual change rates from the northwest ($-39.8 \pm 6.6\%$ or $-0.93 \pm 0.2\%$ a$^{-1}$) aspect, followed by west ($-34.2 \pm 3.7\%$ or $-0.80 \pm 0.1\%$ a$^{-1}$) and east ($-33.6 \pm 5.2\%$

or $-0.78 \pm 0.1\%$ a$^{-1}$) (Fig. 11f). But in terms of absolute area change, glaciers with northern aspect lost maximum area of $-4.6 \pm 0.5$ km$^2$, followed by glaciers with south ($-3.1 \pm 0.3$ km$^2$), west ($-2.8 \pm 0.2$ km$^2$), southeast ($-2.2 \pm 0.2$ km$^2$) and northeast ($-1.8 \pm 0.1$ km$^2$) aspect. Since 1975, the glaciers with mean slope between 15° and 30° exhibit relatively higher retreat rate (Fig. 11b). All these above factors suggest that the glacier's response to climate change is largely controlled by individual glacier morphology, clean ice and debris-cover characteristics, and its associated topographical parameters (e.g.,

size, length, slope, minimum, mean and range elevation, and orientation) on their surface within the study region.

Table 5

Table 6

Figure 11



Figure 12

## 5. Discussion

### 5.1. Comparative evaluation of glacier inventory in the CCW

The present study is an up-to-date glacier inventory (2018) for the CCW in Sikkim Himalaya based on the higher resolution Sentinel 2A MSI satellite image, Google Earth imageries and field based observations. We have identified 74 glaciers with a total area of $44.8 \pm 1.5$ km$^2$ in the CCW for 2018 and this result was compared with some recently published work of the Geological Survey of India (GSI) (Raina and Srivastava, 2008), primarily based on the SOI topographical sheets and aerial photographs. We also compared our results with the glacier outlines complied by ICIMOD (Bajracharya

and Shrestha, 2011) and RGIv6.0 (Mool and Bajracharya, 2003; Nuimura et al., 2015). Table 7 shows that the results of observed total glacier count and their area in this current study (2018) were much different from the ICIMOD (2005 ± 3) database (79 glaciers covering a total area of 45.8 km$^2$) almost after a decade later. In comparison with RGI v6.0 (2000 ± 3) glacier inventory data (90 glaciers covering a total area of 51.1 km$^2$), glacier count was overestimated (7 glaciers), while the total area was underestimated (–6 km$^2$) in the CCW. The major reasons for this different results in the RGI v6.0 database

were due to misinterpretation of single glacier domain (based on slope, aspect and glacier divide) into multiple outlines derived from automated mapping and some glacier boundaries were not delineated mainly in the western part of the watershed (e.g. Lachung Khangse and other unnamed glaciers).

    The GSI glacier inventory presented an overestimated total glacierized area (84 glaciers covering a total area of 80.7 km$^2$) as compared to our estimated area in 1975 (a decade later) using KH-9 image. The possible reasons for this

overestimation were due to topographic map scale limitations (1:50,000) and misinterpretation of glacier outlines due to seasonal snow and debris cover (Raina and Srivastava, 2008; Chand and Sharma, 2015). A previous study by Bhambri and Bolch (2009) also found certain inaccuracies with these topographic maps which were originally derived from aerial photographs acquired during March–June (1964 ± 2), when seasonal snow can create a problem for the correct delineation of glaciers. Thus, many researchers earlier reported similar disadvantages, such as (i) misinterpretation of debris-free and

debris-covered glaciers derived from automated mapping; (ii) misclassification of seasonal snow as glaciers; (iii) temporal and spatial differences of acquired images and mapping period; (iv) division of single ice mass into multiple glaciers domain and vice-versa without checking the local topographical distribution (i.e., slope, orientation, and glacier divide) for delineation process (Bhambri et al., 2011; Chand and Sharma, 2015; Das and Sharma, 2018; Debnath et al., 2019).

Table 7

### 5.2. Topographic controls on area changes

    The spatio-temporal fluctuations of glacier are solely dependent on the local topographic parameters such as glacier size, elevation (minimum, maximum, mean and range), slope and orientation as well as debris cover patterns under existing similar climatic conditions. The high mountain divide (i.e. Kanchengyao–Pauhunri massif) play a significant role in imposing strong topographic influences between the northbound and southbound glacier fluctuations in the region. Glacier

size is also a determining factor for area change. It is evident that the area of size class (<0.5 km$^2$) has gained mainly due the fragmentation and area reduction of existing glaciers in the watershed (Table 5). The overall tendency of glacier area





loss <2 km² reveals that small size glaciers are more sensitive and good indicators of climate change because of their faster response to relatively short-term climate variations (Paul et al., 2004b). We also found a positive correlation (R² = 0.47) between glacier area and absolute glacier area change (Fig. 11a). About 12.1% of the total glaciers with an area of 2.1 km²

have completely disappeared and 1.2% individual glaciers have formed out of fragmentation from the main glacier mass in the study region since 1975.

The debris-cover characteristics play a significant role on the glacier area change in this study as well (Debnath et al., 2019). It is clearly observed from this study that the number of debris free glaciers are dominant in CCW of Sikkim Himalaya. The DF glaciers area changed at a higher rate of –0.27 ± 0.03 km² a⁻¹ than the PDC (–0.12 ± 0.01 km² a⁻¹) and

MDC (–0.02 ± 0.002 km² a⁻¹) glaciers. In this study, PDC glaciers exist on the southern (~25%) slope, whereas such debris-coverage is insignificant for the north faced glaciers, as most of these (~26%) are debris-free and such kind of observations are also found in Bhutan Himalaya by Kääb (2005). These intensive debris supply probably originated from the surrounding steep rock faces of the glaciers in the southern slopes (Frey et al., 2012), which in turn insulate the debris-covered glacier ice to form several dynamic thermokarst features such as depressions and supraglacial ponds in the lower part of the glacier

tongues (Kääb, 2005). It is clearly observed in this study that a relatively gentle slope with lower elevations in the ablation areas can be a favourable place for abundance supply of supraglacial debris.

Glaciers located on the higher elevations above 5300 m.a.s.l on the north face of Kanchengyao–Pauhunri massif (i.e. Gurudongmar and Tista Khangse) is an extension of Trans-Himalaya of Tibetan plateau, an undulating flat surfaces mostly devoid of vegetation as compared to the southern part in the Thangu valley. Our field measurements during December

confirms that mean infrared temperatures of rock (14°C), water (2.7 °C) and grass (11.5°C) on the flat surfaces in the proximity of Gurudongmar lake region (>5150 m) are higher due to more incoming solar radiation than the sloping mountainous terrain near Thangu valley (3900 m) in the south. For example, the mean infrared temperature of rocks, water and grasses near Thangu valley was measured as 4.2°C, 5.1°C and 11.3°C respectively during December. These recorded data reveals the concept of "*plateau flat surface heating*" effect over the higher elevated semi-arid terrain reported by

Brazel and Marcus (1991). Moreover, there is a direct impact on the glacier area change on the northern aspects confirms the significance of incident solar radiation and effects of shadows (Schmidt and Nüsser, 2012). The difference of glacier area loss on the western (2.8 ± 0.2 km² or 0.06 ± 0.005 km² a⁻¹) and eastern (1.1 ± 0.1 km² or 0.06 ± 0.003 km² a⁻¹) aspects during 1975-2018 can be described as more effective melting on the western slopes taking place during the afternoon due to combination of more incident solar radiation and warmer air temperature (Evans, 2006). In addition, the north-facing

glaciers (including northwest, north and northeast) in this region are more susceptible to area loss (7.1 ± 0.7 km² or 0.17 ± 0.02 km² a⁻¹) than the south-facing glaciers (including southeast, south and southwest), which had a total area recession of 6.9 ± or 0.16 ± 0.01 km² a⁻¹. These glacial fluctuations are controlled by the variations of ISM in summer while the mid-latitude westerlies dominates in winters, resulting in a clear seasonality of precipitation which enhanced the glacier melting on the north face of Kanchengyao–Pauhunri massif over the past 43 years (Ali et al., 2018; Benn and Owen, 1998).

**5.3. Regional Comparison with other Himalayan basins**

The present study shows a significant total glacier area loss of 17.9 ± 1.7 km² (28.5 ± 3.6%) with a recession rate of 0.42 ± 0.04 km² a⁻¹ (0.66 ± 0.1% a⁻¹) from 1975 to 2018 in the CCW (Sikkim) Eastern Himalaya. In Sikkim Himalaya, a recent study by Debnath et al. (2019) reported a glacier area loss of 20.7 ± 3.3% (0.51 ± 0.001% a⁻¹) in the Changme Khangpu basin between 1975 and 2016. Racoviteanu et al. (2015) revealed that the glacier area loss of 20.1 ± 8% (0.52 ±



0.2% a⁻¹) in the proximity to Kanchenjunga region in Sikkim side during 1962–2000. Garg et al. (2019) reported a comparatively much lower glacier area loss of $5.4 \pm 0.9$ % ($0.2 \pm 0.04$% a⁻¹) for the 23 randomly selected glaciers in Sikkim (1991-2015) primarily based on medium to higher spatial resolution remote sensing datasets. Similarly, Basnett et al. (2013) had also reported a total loss of $3.3 \pm 0.8$% (~$0.2 \pm 0.1$% a⁻¹) for 39 glaciers between 1989/90 and 2009/10 in Sikkim Himalaya. These published results are comparatively much lower than that of our study as because: (i) shorter time-period

analysis (1990s onwards); (ii) only selected glaciers were mapped for both the case studies. Further east in Bhutan Himalaya, Bajracharya et al. (2014) estimated a comparatively higher glacier area loss of $23.3 \pm 0.9$% ($0.8 \pm 0.03$% a⁻¹) between 1980s and 2010 based on the series of Landsat satellite images.

    In central Himalaya, Bolch et al. (2008) had revealed a planimetric recession of 5.2% (0.12% a⁻¹) in the Khumbu Himalaya of Everest region in Nepal from 1962 to 2005 based on Corona, Landsat TM and ASTER satellite datasets.

Similarly, a glacier loss of 5.9% (0.2% a⁻¹) was reported from the Tamor basin of eastern Nepal during 1970–2000 (Bajracharya and Mool, 2006). However, a significant area loss of $16.9 \pm 6$% ($0.44 \pm 0.2$% a⁻¹) were noticed in the Kanchenjunga region of eastern Nepal (Tamor and Arun basins) between 1962 and 2000 was also reported by Racoviteanu et al. (2015). Bhambri et al. (2011) reported comparatively lower glacier recession of $4.6 \pm 2.8$% ($0.1 \pm 0.1$% a⁻¹) in the Bhagirathi and Saraswati/Alaknanda basin of Garhwal Himalaya for the period of 1968–2006 based on higher resolution

Corona and Cartosat-1 images.

    In context of western Himalaya, Chand and Sharma (2015) reported a total glacier area recession of $4.6 \pm 4.1$% ($0.1 \pm 0.1$% a⁻¹) in Ravi basin of Himachal Pradesh based on high resolution Corona KH-4B, Landsat ETM+ PAN, and WorldView-2 imageries from 1971 to 2010/13. Schmidt and Nüsser (2012) estimated a relative ice cover loss of 14.3% (0.3% a⁻¹) in the Trans-Himalayan Kang Yatze Massif region of Ladakh during 1969–2010 based on high resolution images

(Corona, SPOT 2, World view-1 and Landsat series). Several other studies in the Western Himalaya also revealed a total glacier area loss of $12 \pm 1.5$% in the Sindh and Lidder basins of Kashmir from 1996 to 2014 (Ali et al., 2017); $7.5 \pm 2.2$% ($0.2 \pm 0.1$% a⁻¹) in Jankar chhu watershed of Lahaul and Spiti valley during 1971–2016 (Das and Sharma, 2018); $6.0 \pm 0.02$% ($0.1 \pm 0.0004$% a⁻¹) in Suru sub-basin of Jammu and Kashmir between 1971 and 2017 (Shukla et al., 2020); 8.4% (0.3 % a⁻¹) in Naimona'nyi region of southwest Tibetan Plateau from 1976 to 2003 (Ye et al., 2006). However, a noticeable

deglaciation of $18.1 \pm 4.1$% ($0.5 \pm 0.01$% a⁻¹) in Baspa basin of the Sutlej river between 1976 and 2011 using Landsat data series along with SOI toposheets, Indian remote sensing satellite (IRS) LISS-III was reported by Mir et al. (2017). Again in another study, Kulkarni et al. (2007) reported much higher recession rate of 22% (0.6% a⁻¹) and 21% (0.5% a⁻¹) in Parbati and Chenab sub-basins of Beas river and 19% (0.5% a⁻¹) in the Baspa region of Sutlej river during 1962–2001/04 based on SOI toposheets and LISS III images. These overestimation of mapping glacier outlines using SOI toposheets

could be the possible reasons of much higher recession in the Himachal region of western Himalaya (Bhambri et al., 2011; Chand and Sharma, 2015).

    On the other hand, the glaciers of Karakoram region behave distinctively than the other regions of Hindu Kush Himalaya (HKH) (Hewitt, 1969, 2005; Bahuguna et al., 2014). In another study, Bhambri et al. (2013) reported that 18 glaciers in the upper Shyok valley, Northwest Karakoram showed surge-type behaviour and the study revealed an overall

area gained by $2.2 \pm 56.2$ km² ($+0.1 \pm 3.5$%) during 1973–2011. Hewitt. (2005) mentioned the probable causes of this glacier mass gain in very few areas of central Karakoram Himalaya are due the localized impact of higher elevation and relief as well as a different climatic anomaly involved.

    The quantum of glacier area recession rate in the CCW ($0.42 \pm 0.04$ km² a⁻¹) is similar to that of Changme Khangpu basin

of Sikkim Himalaya (0.45 ± 0.001 km² a⁻¹); Debnath et al. (2019) used higher resolution historical image (e.g. Hexagon)

as well as Sentinel 2A image for glacier delineation. So we can conclude that glaciers in Sikkim (Eastern Himalaya) are retreating at a higher magnitude as compared to the counterparts (i.e. other Himalayan regions) (Bahuguna et al., 2014) and this major shift in the glacier behaviour of Sikkim Himalaya is also noticeable since 2000 in this study (Table 4).

### 5.4. Climate variability and its impact on glacier changes

The glaciers of the Sikkim Himalaya are sensitive to climate warming as a result of ISM and the mid-westerlies

also dominates in winters (Ageta and Higuchi, 1984; Benn and Owen, 1998). The climatic condition of the Karakoram and western Himalaya are different than that of counterpart (Bhambri et al., 2011). The rising temperature contributes to glacier shrinkage over the entire Tibetan Plateau (Fujita and Ageta, 2000; Yao et al., 2012) as well as in the different Himalayan regions (Ageta and Higuchi, 1984; Das and Sharma, 2018; Debnath et al., 2019). Moreover, the response of entire Himalayan glaciers are quite sensitive to precipitation, directly or indirectly through the albedo feedback mechanism

on the short-wave radiation balance (Azam et al., 2018). Table 8 shows the annual and season-wise analysis of MK, Sen's slope (Q) and linear regression tests. The mean annual temperature experienced a significant positive trend (↑) at a rate of 0.249 °C a⁻¹ (0.05 significance level) between 1960–2013. Ironically, winter season experienced the maximum rising trend at the rate of temperature change (0.081 °C a⁻¹), followed by pre-monsoon (0.063 °C a⁻¹), monsoon (0.057 °C a⁻¹) and post-monsoon (0.050 °C a⁻¹) seasons. The annual precipitation has an increasing trend (↑) of 0.639 mm a⁻¹ in the

region. A significant rising trend (↑) of precipitation change rate was observed during pre-monsoon (0.270 mm a⁻¹), followed by winter (0.228 mm a⁻¹; 0.05 significance level) and post-monsoon (0.104 mm a⁻¹) seasons.

But on the other hand, the precipitation change rate during monsoon season experienced a decreasing trend of –0.197 mm a⁻¹. Such significant changes in temperature and precipitation have immense impacts on glacier deglaciation in the investigated region between 1975 and 2018. Fig. 13 confirms that there is a significant increase in the temperature

trend and more or less a static precipitation scenario since 1995 and the rate of glacier area loss also took its pace during 2000-2010 (0.62 ± 0.5 km² a⁻¹) and 2010-2018 (0.77 ± 0.6 km² a⁻¹) timeframes (Table 4). Based on dendrochronology, Yadava et al., (2015) have reported that 1996–2005 as the warmest period for Lachen and Lachung valley (North Sikkim) derived from mean late summer (July–September) temperature reconstruction (AD 1852–2005).

Table 8

Moreover, this significant trend of decreasing precipitation during summer monsoons and warming temperature during all seasons (maximum in winter season) restricts the refreezing of precipitation to form solid ice, thus decrease the degree of accumulation rates than the ablation on the existing glacier surfaces (Benn et al., 2001; Fujita and Ageta, 2000). Similarly, Basnett and Kulkarni (2019) also reported a reduction in snowfall with a declining rate of 2.81 ± 2.02% (−0.3 ± 0.18% a⁻¹) over the Sikkim Himalaya during 2002–2011 caused due to the influence of overall warming climate,

especially the rise in winter minimum temperature and this correlate the finding of the present study. A continuous growth of glacial lakes due to glacier downwasting was also reported in this region of Sikkim Himalaya between 1988 and 2014 (Debnath et al., 2018). Thus, it can be inferred that glacier area shrinkage rates are also dependent on the climatic factors.

Figure 13

### 6. Conclusions

This research presents an integrated watershed based study of glacier change across the CCW in Sikkim Himalaya,



from 1975 to 2018. This glacier analysis comprised of 74 glaciers with a total area of $44.8 \pm 1.5$ km$^2$ including 64 debris free glaciers with an area of $28.4 \pm 1.1$ km$^2$ (63.4% of total glacier area) in 2018. Mean glacier area of the watershed stands at 0.61 km$^2$, with dominance of small-sized glaciers. Our mapping revealed that there has been the glacier area recession of $17.9 \pm 1.7$ km$^2$ in the analysis period, an equivalent to $28.5 \pm 3.6\%$ shrinkage. The overall rate of glacier area loss between 1975–2018 is 0.42 km$^2$ a$^{-1}$ ($0.66 \pm 0.1\%$ a$^{-1}$) in the CCW, consistent with the other basins in Sikkim Himalaya. The present century i.e. 2000–2010 and 2010–2018 have witnessed a higher rate of area shrinkage of $0.62 \pm 0.5$ km$^2$ a$^{-1}$ and $0.77 \pm 0.6$ km$^2$ a$^{-1}$ respectively, as compared to previous timeframes between 1975–1988 ($0.24 \pm 0.4$ km$^2$ a$^{-1}$) and 1988–2000 ($0.20 \pm 0.5$ km$^2$ a$^{-1}$). Hence, the glaciers in the CCW of Sikkim (Eastern Himalaya) are retreating at a higher rate as compared to the other parts (i.e. western and central Himalayas). Larger glaciers (>2 km$^2$) have lost greater area ($7.7 \pm 0.4$ km$^2$ or $-0.18 \pm 0.01$ km$^2$ a$^{-1}$) than the small size classes since 1975, and we suspect that such an anomaly might have been produced by a combined effect of higher solar incidence energy, lower terminus elevation, gentle slopes and associated warmer air temperature. Morphologically, the valley glaciers (SB) have lost the maximum ice cover in the CCW. Overall, the north facing (including northwest, north and northeast) glaciers shrank at a higher rate of $0.17 \pm 0.02$ km$^2$ a$^{-1}$ than in the other aspects. The number of debris free glaciers show a decreasing trend in loss, with a maximum area loss of $11.8 \pm 1.2$ km$^2$ (~$0.27 \pm 0.03$ km$^2$ a$^{-1}$) than the partially and maximum debris-covered glaciers in the CCW. All these factors suggest that the glacier's response to climate change is largely controlled by glacier morphology, surface cover characteristics, and associated local topographical parameters (i.e. size, length, elevation, slope and aspect) within the CCW. Mean annual and seasonal air temperature shows a significant positive trend since 1960s. It appears that the rising trend of temperature during winter season, and declining trend of precipitation in summer season have been the key driving factors for glacier changes in the CCW, however the timeframe of our analysis is much smaller than the response time in the glaciers. In case of continuation of this trend into the future, similar variability in the region will certainly affect the hydroelectric power projects as a result of mobilization of the huge Quaternary deposits stored all along on account of fluctuations in the discharge in this basin.

### 7. Data availability

The temporal datasets developed in this current study for glacier changes in the Chhombo Chhu Watershed of Tista basin, Sikkim Himalaya, India between 1975 and 2018 can now be accessed at Zenodo web portal: http://doi.org/10.5281/zenodo.4457183 (Chowdhury et al., 2021).

### Author contributions

AC has initially designed the idea, collected all primary and secondary data through different sources, gone through the field for detailed investigation and ground truth verification, prepared the final datasets and figures and wrote of the draft manuscript. MD has accompanied during the field survey and laboratory activities. MCS, as a topical expert and joint supervisor has helped in conceptualizing, writing-review and editing. SKD, as an academic supervisor and mentor, has contributed in the entire project administration, supervision, finalizing the structure of the research article and final editing. All the authors have significantly contributed to the final form of this manuscript.

### Competing interests

The authors declare that they have no conflict of interest.



**Acknowledgments**

We would like to thank the Forest Department, Govt. of Sikkim for issuing the permit for field visits. This study was funded by University Grant Commission, New Delhi (36051/ NET–DEC. 2012) between 2016 and 2018 and, ICSSR Doctoral Fellowship (2019-2020). The authors are also obliged to North-Eastern Hill University (Shillong) and Jawaharlal Nehru University (New Delhi) for providing the research and laboratory facilities. We also thank USGS for providing declassified Hexagon (KH-9), Landsat TM/ETM+, Sentinel 2A MSI and SRTM DEM data at no cost. Special thanks to Mr. Tejashi Roy (Jawaharlal Nehru University, New Delhi) who helped in the field measurements during November 2018.

Sincere thanks to Dr. Ian Harris, University of East Anglia (UK) for providing the CRU station data and valuable suggestions. We greatly appreciate the efforts of anonymous reviewers for constructive suggestion to improve the content and quality of our paper.

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





Table 1. Different datasets used in this current study for the analysis and its characteristics

| Maps/Sensors | Product/Scene ID | Acquisition date | Spatial resolution (m) | Temporal resolution (days) |
|---|---|---|---|---|
| Hexagon (KH9-11) | DZB1211-500057L037001_37 | 1975-12-20 | PAN (4 m) | – |
| Landsat 5 (TM) | LT51390411988336BKT00 | 1988-12-01 | VIS + MIR (30 m) | 16 |
| Landsat 7 (ETM+) | LE71390412000313SGS00 | 2000-11-08 | PAN (15 m); VIS + MIR (30 m) | 16 |
| Landsat 5 (TM) | LT51390412010348KHC00 | 2010-12-14 | VIS + MIR (30 m) | 16 |
| Sentinel (2A-MSI) | L1C_TL_EPAE_20181126T073934_A017905_T45RXM L1C_TL_EPAE_20181126T073934_A017905_T45RXL | 2018-11-26 | VIS (10 m); SWIR (20 m) | 5 |
| GE platform | – | – | 1.65 to 2.62 m | – |
| SRTM DEM | – | 2000 | 30 m | – |

Note: KH, KeyHole; GE, Google Earth; SRTM DEM, Shuttle Radar Topography Mission digital elevation model; TM, Thematic Mapper; ETM+, Enhanced Thematic Mapper Plus; MSI, multispectral instruments; PAN, panchromatic; VIS, visible; MIR, mid-infrared; SWIR, shortwave infrared.


Table 2. Topographic parameters according to different size classes (2018) for the CCW based on Sentinel 2A and SRTM
DEM.

| Parameters | Size class (km²) | | | | | |
|---|---|---|---|---|---|---|
| | <0.5 | 0.5–1 | 1–1.5 | 1.5–2 | >2 | CCW |
| Number of glaciers | 51 | 13 | 4 | 1 | 5 | 74 |
| Mean size (km²) | 0.21 | 0.68 | 1.22 | 1.86 | 3.74 | 0.61 |
| Total Glaciated area (km²) | 10.55 (24%) | 8.84 (20%) | 4.86 (11%) | 1.86 (4%) | 18.69 (42%) | 44.8 |
| Average elevation mean (m) | 5526 | 5809 | 5677 | 5670 | 5700 | 5598 |
| Average elevation range (m) | 357 | 600 | 1084 | 751 | 1334 | 511 |
| Minimum elevation (m) | 4688 | 5053 | 4770 | 5363 | 4851 | 4688 |
| Maximum elevation (m) | 6574 | 6911 | 6910 | 6114 | 6758 | 6911 |
| Mean slope (°) | 27 | 24 | 30 | 20 | 24 | 26 |
| Mean aspect (°) | SE (151) | SE (153) | S (169) | N (22) | SE (133) | SE (150) |
| DF glacier area (km²) | 9.87 (94%) | 7.56 (86%) | 2.5 (51%) | 1.86 (100%) | 6.66 (36%) | 28.4 (64%) |
| PDC glacier area (km²) | 0.38 (4%) | - | 2.36 (49%) | - | 12.03 (64%) | 14.8 (33%) |
| MDC glacier area (km²) | 0.3 (3%) | 1.28 (14%) | - | - | - | 1.6 (4%) |

DF, Debris Free; PDC, Partially Debris Covered; MDC, Maximum Debris Covered




Table 3. Glacier parameters according to morphological types (2018) for the Chhombo Chhu Watershed based on Sentinel 2A and SRTM DEM.

| Parameters | Morphological types | | | | | |
|---|---|---|---|---|---|---|
| | Valley (CB) | Valley (SB) | Mountain (SB) | Cirque | Niche | Glacieret (SF & IA) |
| Number of glaciers | 2 | 17 | 21 | 9 | 10 | 15 |
| Mean size (km$^2$) | 3.2 | 1.1 | 0.6 | 0.3 | 0.2 | 0.2 |
| Total glaciated area (km$^2$) | 6.4 (14%) | 18.0 (40%) | 13.3 (30%) | 2.5 (6%) | 1.7 (4%) | 2.9 (6%) |
| Average elevation mean (m a.s.l.) | 5743 | 5473 | 5572 | 5430 | 5705 | 5785 |
| Average elevation range (m) | 1210 | 662 | 628 | 388 | 365 | 251 |
| Minimum elevation (m a.s.l.) | 4957 | 4688 | 4770 | 4825 | 5178 | 4924 |
| Maximum elevation (m a.s.l.) | 6474 | 6758 | 6910 | 6178 | 6683 | 6911 |
| Mean slope (°) | 24 | 23 | 26 | 27 | 31 | 28 |
| Mean aspect (°) | SE (154) | SE (133) | S (162) | S (168) | SE (136) | SE (148) |
| DF glacier area (km$^2$) | – | 13.5 (75%) | 7.8 (59%) | 2.5 (100%) | 1.7 (100%) | 2.9 (100%) |
| PDC glacier area (km$^2$) | 6.4 (100%) | 2.9 (16%) | 5.5 (41%) | – | – | – |
| MDC glacier area (km$^2$) | – | 1.6 (9%) | – | – | – | – |

CB, Compound basin; SB, Simple basin; SF, Snow fields; IA, Ice Aprons; DF, Debris Free; PDC, Partially Debris Covered; MDC, Maximum Debris Covered

Table 4. Glacier area dynamics in total area in the Chhombo Chhu watershed, Sikkim Himalaya (1975–2018)

| Years | No. | Glacier area (km$^2$) | | Periods | Area change | | Rate of area change | |
|---|---|---|---|---|---|---|---|---|
| | | Mean | Total area | | km$^2$ | % | km$^2$ a$^{-1}$ | % a$^{-1}$ |
| 1975 | 83 | 0.75 | 62.6 ± 0.7 | 1975–1988 | –3.1 ± 5.3 | –5.0 ± 9.0 | –0.24 ± 0.4 | –0.39 ± 0.7 |
| 1988 | 83 | 0.72 | 59.5 ± 5.3 | 1988–2000 | –2.4 ± 5.9 | –4.0 ± 10.0 | –0.20 ± 0.5 | –0.33 ± 0.8 |
| 2000 | 83 | 0.69 | 57.1 ± 2.6 | 2000–2010 | –6.2 ± 5.5 | –10.8 ± 10.5 | –0.62 ± 0.5 | –1.08 ± 1.0 |
| 2010 | 80 | 0.64 | 51.0 ± 4.8 | 2010–2018 | –6.2 ± 5.0 | –12.1 ± 10.0 | –0.77 ± 0.6 | –1.51 ± 1.3 |
| 2018 | 74 | 0.61 | 44.8 ± 1.5 | 1975–2018 | –17.9 ± 1.7 | –28.5 ± 3.6 | –0.42 ± 0.04 | –0.66 ± 0.1 |

Table 5. Glacier area changes as per size class (1975–2018) in the CCW

| Size class | 1975 | | 2018 | | 1975–2018 | | | |
|---|---|---|---|---|---|---|---|---|
| | | | | | Area change | | Rate of area change | |
| km$^2$ | No. | km$^2$ | No. | km$^2$ | km$^2$ | % | km$^2$ a$^{-1}$ | % a$^{-1}$ |
| <0.5 | 48 | 10.3 ± 0.2 | 51 | 10.5 ± 0.6 | +0.3 ± 0.6 | +2.6 ± 6.1 | +0.01 ± 0.01 | +0.06 ± 0.1 |
| 0.5–1 | 21 | 14.5 ± 0.2 | 13 | 8.8 ± 0.3 | –5.6 ± 0.4 | –39.0 ± 3.8 | –0.13 ± 0.01 | –0.91 ± 0.1 |
| 1–1.5 | 6 | 8.3 ± 0.1 | 4 | 4.9 ± 0.2 | –3.4 ± 0.2 | –41.5 ± 3.4 | –0.08 ± 0.004 | –0.96 ± 0.1 |
| 1.5–2 | 2 | 3.2 ± 0.04 | 1 | 1.9 ± 0.04 | –1.4 ± 0.1 | –42.3 ± 2.5 | –0.03 ± 0.001 | –0.98 ± 0.1 |
| >2 | 6 | 26.4 ± 0.2 | 5 | 18.7 ± 0.4 | –7.7 ± 0.4 | –29.1 ± 2.2 | –0.18 ± 0.01 | –0.68 ± 0.1 |
| Total | 83 | 62.6 ± 0.7 | 74 | 44.8 ± 1.5 | –17.9 ± 1.7 | –28.5 ± 3.6 | –0.42 ± 0.04 | –0.66 ± 0.1 |



Table 6. Glacier area changes according to morphological types (1975–2018) in the CCW

| Morphological types | 1975 | | 2018 | | 1975–2018 | | | |
|---|---|---|---|---|---|---|---|---|
| | | | | | Area change | | Rate of area change | |
| | No | Area (km²) | No. | Area (km²) | km² | % | km² a⁻¹ | % a⁻¹ |
| Valley (CB) | 2 | 8.3 ± 0.1 | 2 | 6.4 ± 0.1 | −1.9 ± 0.2 | −22.7 ± 2.3 | −0.04 ± 0.004 | −0.53 ± 0.1 |
| Valley (SB) | 20 | 26.4 ± 0.3 | 17 | 18.0 ± 0.5 | −8.4 ± 0.6 | −31.7 ± 3.0 | −0.19 ± 0.01 | −0.74 ± 0.1 |
| Mountain (SB) | 22 | 18.4 ± 0.2 | 21 | 13.3 ± 0.5 | −5.1 ± 0.6 | −27.6 ± 4.0 | −0.12 ± 0.01 | −0.64 ± 0.1 |
| Cirque | 10 | 3.4 ± 0.1 | 9 | 2.5 ± 0.1 | −0.9 ± 0.1 | −25.3 ± 5.0 | −0.02 ± 0.003 | −0.59 ± 0.1 |
| Niche | 11 | 2.2 ± 0.04 | 10 | 1.7 ± 0.1 | −0.5 ± 0.1 | −23.2 ± 5.9 | −0.01 ± 0.002 | −0.54 ± 0.1 |
| Glacieret (SF & IA) | 18 | 4.1 ± 0.1 | 15 | 2.9 ± 0.1 | −1.2 ± 0.2 | −29.2 ± 5.1 | −0.03 ± 0.004 | −0.68 ± 0.1 |
| Total | 83 | 62.6 ± 0.7 | 74 | 44.8 ± 1.5 | −17.9 ± 1.7 | −28.5 ± 3.6 | −0.42 ± 0.04 | −0.66 ± 0.1 |

Table 7. Comparison of glacier inventory within the study region (CCW).

| Sl. No. | Glacier Inventory | No. of glaciers | Total Area (km²) | Data used and year | References |
|---|---|---|---|---|---|
| 1. | GSI (2008) | 84 | 80.7 | SOI Toposheets and aerial photographs (1964 ± 2) | *(Raina and Srivastava, 2008)* |
| 2. | ICIMOD (2011) | 79 | 45.8 | Landsat 7 ETM + (2005 ± 3) and SRTM DEM | *(Bajracharya and Shrestha, 2011)* |
| 3. | RGI v6.0 (2017) | 90 | 51.1 | Landsat 7 ETM + (2000 ± 3) | *(Mool and Bajracharya, 2003; Nuimura et al., 2015)* |
| 4. | Present Study (2018) | 74 | 44.8 ± 1.5 | Sentinel 2A MSI (2018) and SRTM DEM | *This study* |


Table 8. Statistical results of Mann–Kendall test for trend analysis of long-term annual and seasonal temperature and precipitation over the period 1960–2013. Sen's Slope (Q) analysis shows the rate of change in temperature (°C a⁻¹) and

precipitation (mm a⁻¹).

| Seasons | Temperature | | | | Precipitation | | | |
|---|---|---|---|---|---|---|---|---|
| | $Z_{MK}$ | Linear equation | Q value | Trend | $Z_{MK}$ | Linear equation | Q value | Trend |
| | | | (°C a⁻¹) | | | | (mm a⁻¹) | |
| Spring (Pre-monsoon) | 3.58* | y = 0.0663x - 131.73 | 0.063* | ↑ | 1.54 | y = 0.3025x - 527.77 | 0.270 | ↑ |
| Summer (Monsoon) | 3.15* | y = 0.0569x - 84.591 | 0.057* | ↑ | −0.37 | y = -0.1222x + 550.71 | −0.197 | ↓ |
| Autumn (Post-monsoon) | 2.75* | y = 0.0577x - 117.59 | 0.050* | ↑ | 0.86 | y = -0.0038x + 37.453 | 0.104 | ↑ |
| Winter | 2.26* | y = 0.0836x - 188.22 | 0.081* | ↑ | 2.75* | y = 0.2342x - 451.38 | 0.228* | ↑ |
| Annual | 2.91* | y = 0.2602x - 513.61 | 0.249* | ↑ | 1.33 | y = 0.4138x - 397.23 | 0.639 | ↑ |

where, Spring (Mar-May), Summer (Jun-Sep), Autumn (Oct-Nov), Winter (Dec-Feb); (↑), increasing trend; (↓), decreasing trend; * statistically significant at 0.05 alpha level of significance or 95% confidence level.



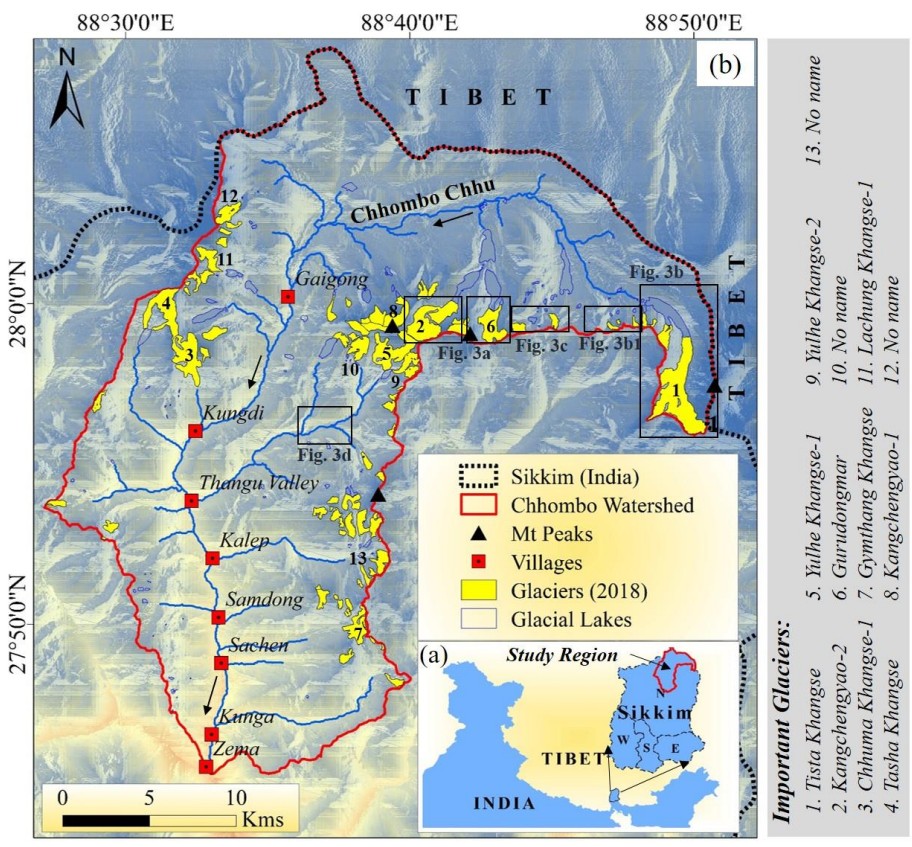

Figure 1. (a) Location of the Chhombo Chhu watershed in the Eastern Himalaya. The watershed boundary is marked in red in the inset within Sikkim, (b) Glacier and glacial lake distribution in the CCW in the Sikkim Himalaya. Inset in black boxes represent the field measurement sites in 2017–2018. The base map used here is SRTM DEM.





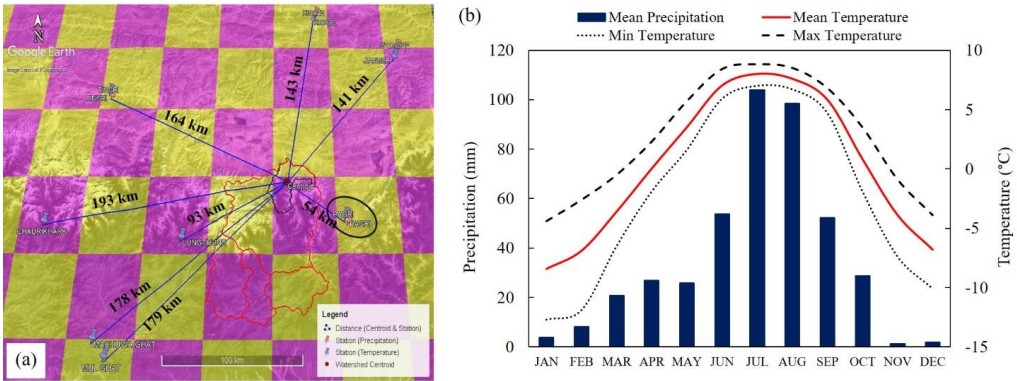

Figure 2. (a) Location of meteorological stations overlaid on Google Earth platform and its respective distance (km) to the
centroid of the CCW. The red and blue placemarks represent the individual stations for precipitation and temperature data
presnted in this study. Black oval inset represent the Pagri Meteorological Station (PMS). Pink and yellow checker boxes
represent the 0.5°×0.5° grided cells of CRU data (b) Climograph of the CCW, Sikkim Himalaya, representing the mean
monthly temperature and precipitation data from 1960 to 2013; Data Source: (http://www.cru.uea.ac.uk/data).






Figure 3. Field photographs (2017-18) showing the terminus characteristics of some important glaciers in the study region

(see Fig. 1 for location). IF, icefall; CI, clean ice; PDCI, partially debris-covered ice; MCF, Mountain Cliff Face; OLM, old lateral moraine; NLM, new lateral moraine; UCLM, unconsolidated Lateral Moraine; PGL, proglacial lakes; GT, glacial tarn; TK, Tista Khangse. The red oval inset surrounding person represent the scale of the image. (All Photo courtesy: Chowdhury, A. 2017-18)

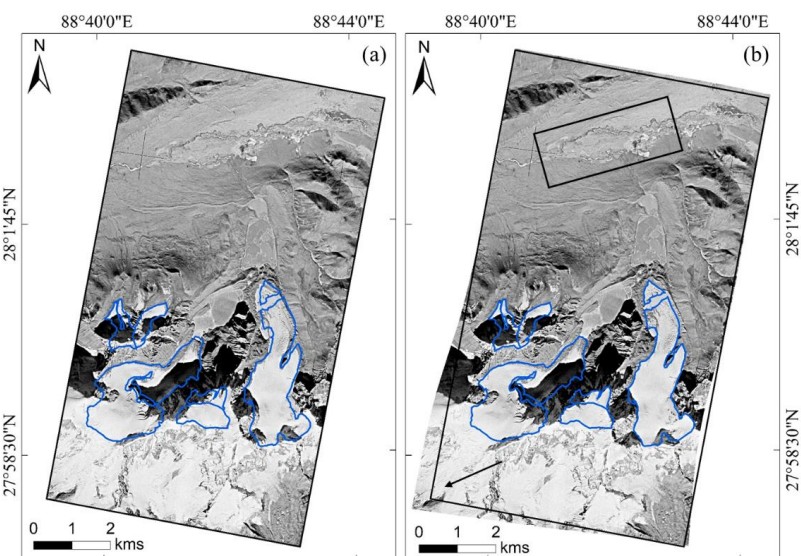

Figure 4. Methods adopted for glacier mapping: (a) Subset KH-9 image near Gurudongmar region before perfect geometric rectification using Spline Transformation Method (STM), (b) Distortion fields of the same KH-9 image along the mountain ridges after geometric rectification. Note: The difference of less distortion were clearly noticed near the flat areas of braided glacial streams (rectangle) in the north of Gurudongmar Tso than the Kangchengyao massif (arrow) in the south.


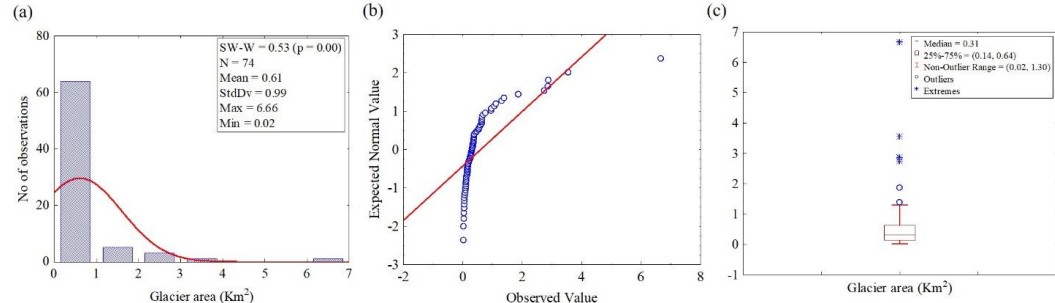

Figure 5. Distribution of glaciers in the CCW of Upper Tista basin, Sikkim Himalaya (2018) (a) histogram of glacier areal distribution; (b) normal Q-Q plot of glacier area (c) box-whisker plot of glacier surface area.


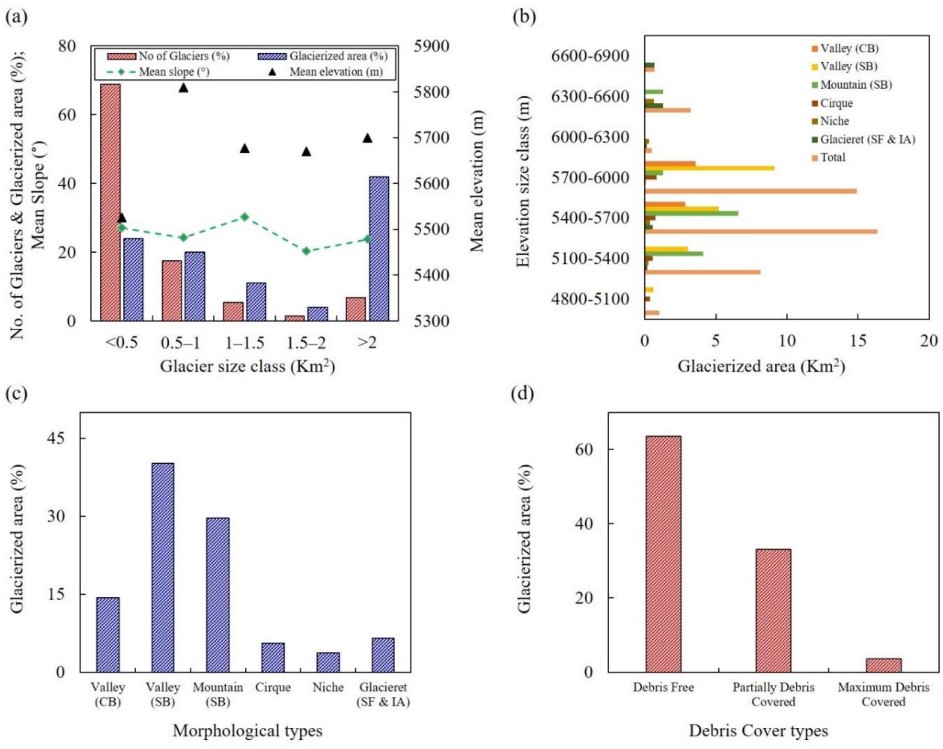

Figure 6. Distribution of (a) number of glaciers, glacier area, mean slope and mean elevation as per glacier size classes, (b) elevation size classes of morphological types in relation to glacier area, (c) Frequency distribution (in percentage) of glacier morphological types and, (d) debris cover types.




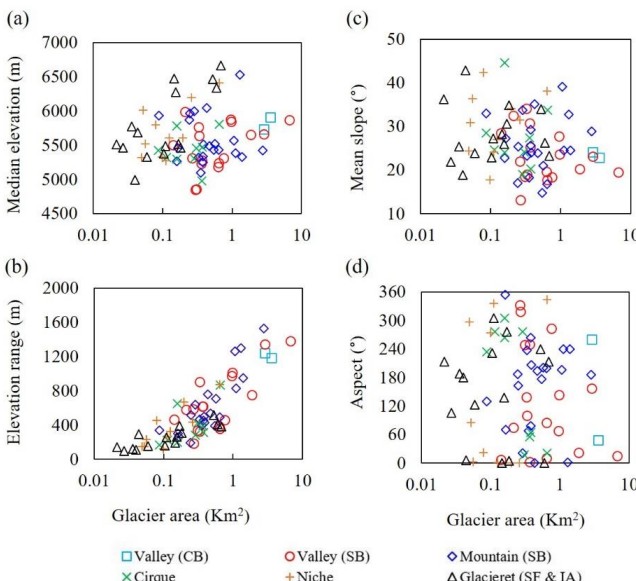

Figure 7. Scatter plots showing relationships between different morphological types of glacier area (km²) and (a) mean elevation (m); (b) mean slope (°); (c) elevation range (m); and (d) aspect (°). Glacier characteristics are derived from Sentinel 2A MSI and SRTM DEM.


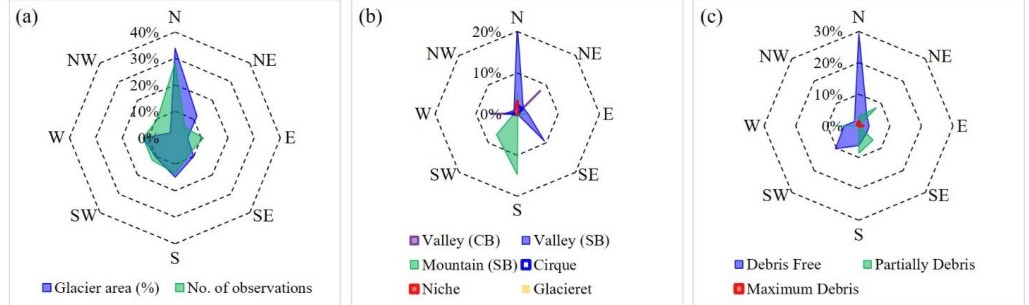

Figure 8. Distribution of the glaciers (74) according to aspect in the CCW based on Sentinel 2A MSI (26 November 2018) and SRTM DEM. (a) Glacier frequency distribution and total glacier area in percent; (b) Glacier area (%) according to morphological types; (c) Glacier area (%) in relation to debris covered types. All the data are in percentage (%) and on average, the glaciers in this watershed are predominantly oriented towards N (0°) and followed by S (180°).



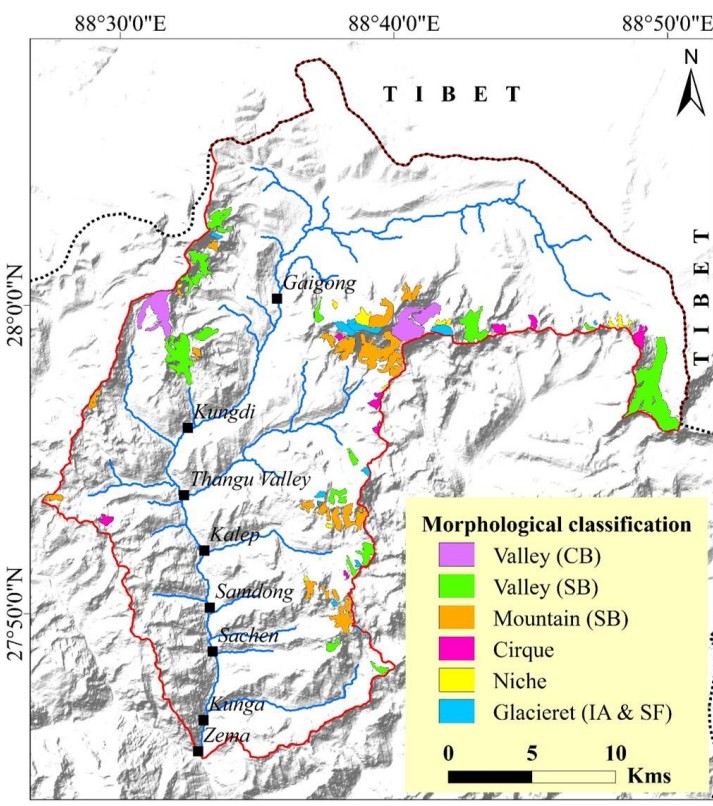

Figure 9. Morphological classification of glaciers in the CCW, Sikkim Himalaya. The base map is the Hill Shaded Relief map of SRTM DEM. (For interpretation of the references to colour in this figure legend, the reader is referred to the web
version of this article.)






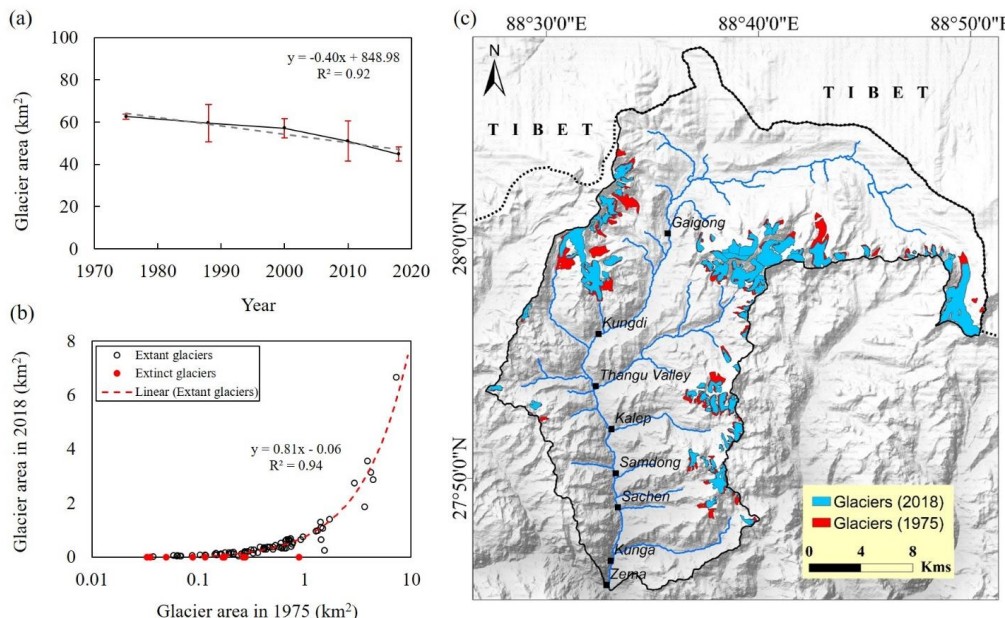

Figure 10. Glacier surface areal change in the Chhombo Chhu watershed: (a) Total glacier area change (dash grey line is
exponential best-fit relationship with associated equation and $R^2$ value). Error bars are the uncertainty (%) of respective
years estimated using the Eq. (1); (b) Scatter plot of glacier area of 1975 and 2018. Red bullets in the graph indicates the
10 extinct glaciers during the phase of 1975–2018; (c) Glacier retreat (1975–2018) map over a timeframe of 43 years of
the study region. The base map is the Hill Shaded Relief map of SRTM DEM. (For interpretation of the references to colour
in this figure legend, the reader is referred to the web version of this article.)




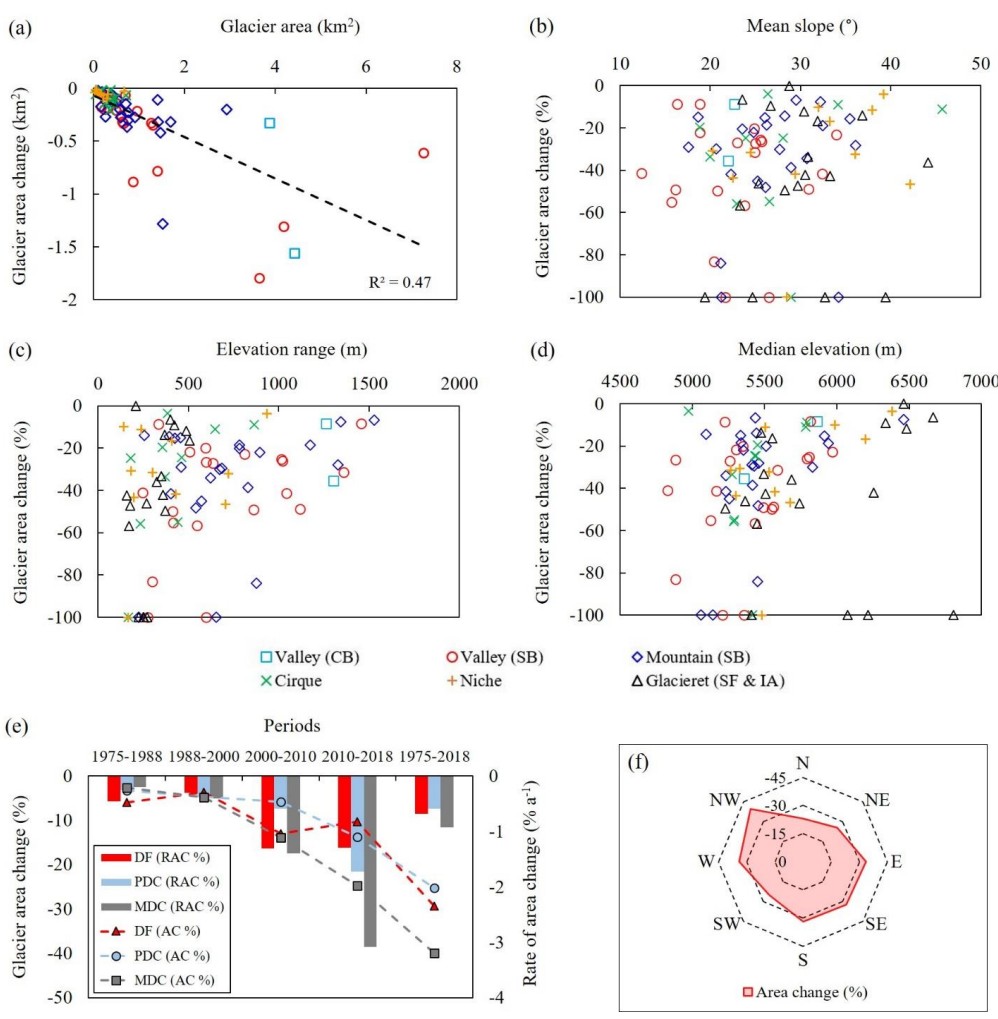


Figure 11. Scatter plots showing the relationships between glacier area changes during 1975–2018 on (a) glacier area (km$^2$);
(b) mean slope (°); (c) elevation range (m); (d) median elevation (m) (e) different periods of debris cover types; and (f)
aspects (°).

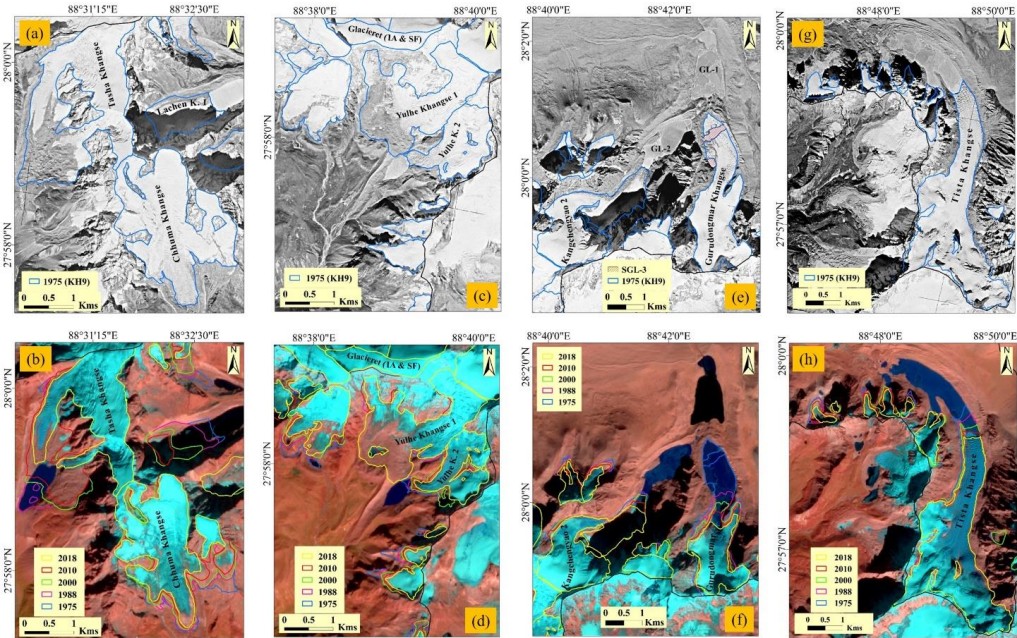

Figure 12. Glacier areal change (East to West direction from the right) in the Chhombo Chhu watershed from 1975 to 2018: (a,c,e,g) Glacier outlines (in blue) drawn on the rectified subsets of Declassified Hexagon (KH9) image (20 December 1975) based on the spline transformation method with similar-year glacier outline; (b,d,f,h) Spatio-temporal areal change of different glaciers in the watershed are overlaid on the Sentinel 2A MSI (2018) imagery.

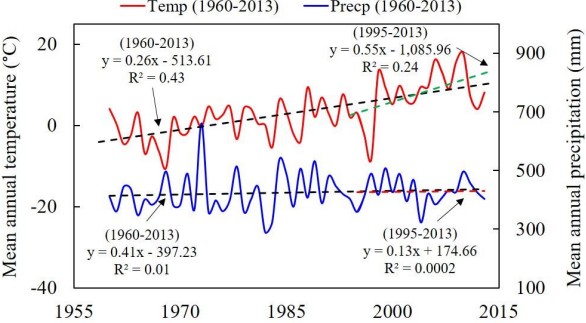

Figure 13. Trends in mean annual temperature (°C) and precipitation (mm) between 1960 and 2013 in the CCW based on the Pagri Meteorological Station (PMS). Data source: (http://www.cru.uea.ac.uk/data).