# Peer review of "Glacier Changes in the Chhombo Chhu Watershed (CCW) of Tista basin between 1975 and 2018, Sikkim Himalaya, India"

_Earth System Science Data, 2020_

## Author Comment (AC2)

The authors express their gratitude to the editorial team and the reviewers for their insightful feedback and suggestions to upgrade our manuscript. The detailed specific responses to the reviewer's comments are given below. The responses of the comments and the revisions in the manuscript are given in blue colour.

**Reviewer #2: 05 April 2021**

Review on the manuscript entitle "Glacier Changes in the Chhombo Chhu Watershed of Tista basin between 1975 and 2018, Sikkim Himalaya, India".

**Comments to the authors**

After several read, I find this work very informative, especially the illustration (figures) and data analysed are impressive. This shows the degree of data processing and visibility. One way it may attracts many readers to follow, on the other hand the study has focused only a small catchment containing only 74 glaciers (2018). Considering, the V order basin of the Himalaya, there are >100 of V order basin and 10 times of that the glacierized catchment. Since, remote sensing techniques are well known for its spatial coverage. I would suppose to have this study at least to a basin level with the same efforts.

Thanks for your valuable suggestions.

**General comments**

Authors could be clearer about what readers can learn from the data about the temporal changes in the length of the glaciers. The following are comments elaborate on this point.

**General response:** The present study deals only with the glaciers area changes in the watershed. Much of the loss in glacier cover in the frontal sections were observed, therefore the length can be deciphered from the data provided in the Zenodo portal: http://doi.org/10.5281/zenodo.4457183 (Chowdhury et al., 2021).

**Comment:** Basically, this work has focused on the inventory of the glaciers in the Chhombo Chhu Watershed and tried to explain the enhanced retreat rate during recent with metrological data. Climate variability and its impact on glacier changes show no convincing agreement in terms of temporal variations. This must be worth discussing.

**Response:** This sentence has been incorporated in the text (Page 14, Line 482-485) "Since the glacier response time is of many decades and centuries, the contemporary data may not have any perceptible reflections immediately but over the decades, it is also possible that the present trend of loss in the glaciers mass may be a response to climatic conditions that existed a few centuries ago, given the mass transfer with respect to velocity, ranging from ~5-25 m a$^{-1}$ in most the Himalayan glaciers."

**Comment:** Similarly: In the last sentence of the abstract, you state that this inventory will provide valuable information for planning the water resources in the Sikkim state of Eastern Himalaya. But such information of temporal changes, planning and execution is not discussed in the text.

**Response:** This sentence is deleted from the text because really it does not have any relation with the present work.

**Comment:** Another thing that was lacking in the manuscript is proper terminology and the literature review. For instance, (PDC) (MDC), mountain (SB) valley (CB) use of such abbreviation is hard to follow as it is newly added and several times used in the manuscript. Use, debris-covered, fully debris-covered, partially debris-covered, mountain glacier, valley glaciers or else. Length change, retreat rate, recessional rate, glacier area loss or glacierized area loss. Fix these words and follow consistently throughout the manuscript.

**Response:** Thank you very much for this valuable suggestion. All of those abbreviations have been replaced with full terminologies throughout the manuscript.

**Detailed comments**

**Comment:** Abstract: Debris free: use clean ice or clean glacier instead debris free.
**Response:** It is replaced with clean glacier throughout the manuscript.

**Comment:** In fact, it should be mentioned in the method that how you categorized the Clean 'C' type and debris-covered 'D' type glaciers. Partially debris-covered: being quantitative mentioned the % for partial and maximum debris-cover.

**Response**: New Lines 159-1160: It is incorporated in the method section. "We have considered the glacier with more than 50% debris-cover as termed as maximum debris-covered, while less than 50% is called partially debris-covered for a qualitative differentiation and debris-free glacier has been termed as clean glacier."

**Comment:** L18-22: Use quantitative results in the abstract, rather to provide general statement.

**Response:** New Line 19: Suggestions have been incorporated in the abstract.

**Comment:** L29: This sentence is not understandable, or required proper validation such as referencing.

**Response:** New Line 22: This sentence has been reframed.

**Comment:** L30: How and what is the response indicators, is it response time of the glacier? If so then try to explain response indication glaciers in prior sentence.

**Response:** New Line 26: This sentence is added before that sentence to make it more clear: 'Response of negative mass balance is shown in the frontal area as recession supported by identifiable landforms'.

**Comment:** L31: Such comparison in the sentence actually raising curiosity in readers that how large is the difference, is it two-fold or comparably similar specific loss.

**Response:** New Lines 29-32: Suggestions have been incorporated in the text. "Most of the Himalayan glaciers are losing mass at different rates, such as Khatling glacier in the Garhwal Himalayas and Zemu glacier in the Sikkim Himalayas are retreating at the rate of $88 \pm 0.3$ m $a^{-1}$ and 9.1 m $a^{-1}$ respectively (Raj et al., 2017; Rashid and Majeed, 2020)".

**Comment:** L33-34: referencing is more than what was mentioned in the text.

**Response**: New Line 32: Keeping only the first references of 2011 rest of the references are deleted.

**Comment:** L42: Delete 'However'.

**Response:** Deleted and changed the structure of the sentence.

**Comment:** L44: This sentence should come first.

**Response**: New Lines 40-41: The second sentence is shifted at the beginning of the paragraph, as suggested.

**Comment:** L45: Any of the bigger statement need proper citation.

**Response:** New Line 43-45: These sentences have been reframed and suggestion have been incorporated.

**Comment:** L46: sediment related concern needs proper background, why this brings in context to the present study.

**Response:** This sentence is deleted. Thank you very much.

**Comment:** L52-53: It does not make any sense, even cited article not intended to say this statement.

**Response:** This sentence is deleted

**Comment:** L54-56: If this is the case for present study, then all basins are somehow related to the same problems. Here need to state the fundamental of present study by answering how and why.

**Response:** Suggestions have been incorporated, New Lines 49-51 : "The Tista being a trans-boundary river, sustaining a sizable population both in and at the distal reaches of the mountain (Rudra, 2018; Rahaman and Mamun, 2020), our change analysis of the cryospheric parameter would help in long-term planning along the river course, given the current trend."

**Comment:** L58: Delete the whole line. Avoid to use local-level, try to use small scale, regional level, basin level, catchment level etc.

**Response:** This sentence is deleted

**Comment:** L73: Delete, on glaciers since historical past.

**Response:** Deleted this part from the text (New Line 65).

**Comment:** L86-88: Here describe about the climate of the area, not that ISM plays role in glacier fluctuation. Such data article actually needs a greater detail of climate of the study area.

**Response**: New Lines 78-82: Suggestion has been incorporated and the statement has been rewritten by describing the climate of the entire region with additional citation (Murari et al. 2014; Kumar et al., 2020). Moreover, New Lines 84-87 have also been revised.

**Comment:** Suggestion on referencing: the two article Debnath, 2018 and 2019, which were cited 18 times in the manuscript, brings special attention to read both of them, while there were different objectives described, here it was cited for different context. Authors, need to be careful with statements and article citation.

**Response:** Citations of these two articles have been reduced from 18 to 4. Most of the irrelevant sentences related to these references have also been deleted.

**Comment:** L170: please acknowledge the university or institute having ArcGIS 10.2.2 license.

**Response:** We have ArcGIS 10.1 software license at Jawaharlal Nehru University, New Delhi and it is incorporated in the text in acknowledgement. Thanks for pointing it out.

**Comment:** Section 3.6. L201: Provide the aspects of extensive field measurements, means the instrument used or it is visual interpretation and shown with figure 3.

**Response:** The details on the instrument used and reason behind the application has been discussed in details about the nature of filed activities done during the field survey (New Lines 195-197 and 203-206).

"Land surface temperature on the northern and southern part of the watershed for bare rocks, grasses, moraine, water etc. has been measured using the Fluke infrared thermometer. The Digi-Schmidt hammer 2000 has been used to measure the relative hardness of boulders that infers about the relative weathering and relative time of glacier recession from the places."

**Comment:** L206: delete 'dangerous'

**Response:** Deleted

**Comment:** L208: If it is benchmark glacial lakes, there must be name of it.

**Response:** Suggestions have been incorporated in the text (New Line 203).

**Comment:** L212: For minimum size threshold, as per RGI they consider 0.01 km$^2$ the minimum size, but recent inventories have considered 0.02 km$^2$ (Frey et al., 2012; Chand and Sharma, 2015). This information should come in method section with reference.

**Response:** Included this sentence in the last part of the first paragraph under point 3.3 (glacial inventory mapping) (New Lines 154-155)

**Comment:** L220-225: This will be better while comparing with other studies in discussion section, so shift it.

Response:  Shifted in the point 5.3 (Regional Comparison with other Himalayan basins)

**Comment:** L286-289: This can be shift to discussion section. And delete Fig. 10c showing the retreat map of the study area (1975–2018).

Response:  Shifted in the point 5.1 last sentence (Comparative evaluation of glacier inventory in the CCW- New Lines 350-351). The caption of Fig. 10c is revised and cited in New Line 275.

**Comment:** L290: Describe the results and then followed by the referred figure.

Response: Figure number has been shifted after results (New Line 282)

**Comment:** General: L300: Better term is clean glacier or 'C' type, not debris free DF, and please used full word for terminology having definitions. It is interesting that having enhanced retreat rate of the glaciers in this catchment, still have higher number of clean glaciers. Ok what is the criteria set for clean glaciers in this study? PDC: same here, how much debris cover (in % either for the total glacier area or for total ablation area) was categorized for partial debris-covered glaciers.

**Response:** Suggestions have been incorporated and the sentence is revised according to the suggestion (P9-L314). All of those abbreviations have been replaced with full terminologies throughout the manuscript. The criteria are based on qualitative differentiation which is already mentioned in the method section (New Lines 159-160)

**Comment:** L305: "mainly due to lower terminus elevation" This is not the reason for recession, it is a mechanism that glaciers having longer length and lower terminus position has receded at higher rate due to higher negative mass balance at the lower elevation.

**Response:**  Deleted "mainly due to lower terminus elevation" and added one sentence 'Due to the glaciers having a longer length and lower terminus position, it has receded at a higher rate due to higher negative mass balance at the lower elevation." (New Lines 296-297)

**Comment:** L313-314: This is the process observed but the deferential processes is something else, which I described in the above comment. So rather, you may show the observation instead the reasons.

**Response:** The sentence is changed like this "It is found that the maximum glacier area loss is taking place at the lower terminus elevation and comparatively lesser slope of large valley (simple basin) glacier than the other morphological types." (New Lines 304-305)

**Comment:** 5.1 CCW: Chhombo Chhu Watershed

**Response:** Corrected it (New Line 325)

**Comment:** 335-345: For this comparison between inventories of glaciers, providing detail only in Table will not justify. For valid comparison and stating the discrepancies of other inventory you might have to provide a figure with glacier outline. I think it is available for all datasets.

**Response:** Suggestions have been incorporated (New Line 338)

[Figure]

Figure 13. Glacier inventory: (a) overlay of glacier outlines of different inventory on Landsat 7 ETM+ 8 November 2000; (b) Slope map (in °) showing the misinterpretation of glacier boundary and ice/sub-watershed divides. Dotted black lines represent the watershed divide.

**Comment:** 348-353: First, Authors have selected wrong dataset for GSI inventory, so check with Sangewar and Shukla, 2009. Second, that was the era, when Geological Survey of India has make larger efforts to provide a detail study of Indian Himalayan glaciers. So, I suggest to follow their glaciological terminology and the best out of the old literature. Avoid to say misinterpretation, rather you can provide uncertainties.

**Response:** New Line 329: Error has been corrected as we have now considered the published Glacier Atlas of India by Raina and Srivastava (2008) only. Suggestions (such as full terminologies for morphological types) are well taken and incorporated accordingly throughout the text.

**Comment:** 384: December: provide years and number of days with time of measurement. The surface temperature has diurnal variability. So comparing morning and evening may lead to misinterpretation.

**Response:** Suggestions have been incorporated (New Line 384).

**Comment:** L394-397: Though, the difference is very much close, still authors needs to discuss the reason in here, rather to provide results.

**Response**: Suggestions have been incorporated. The probable controlling factors have been identified and elaborated in the manuscript (New Lines 377-397).

The difference of glacier area loss on the western ($2.8 \pm 0.2$ km$^2$ or $0.06 \pm 0.005$ km$^2$ a$^{-1}$) and eastern ($1.1 \pm 0.1$ km$^2$ or $0.06 \pm 0.003$ km$^2$ a$^{-1}$) aspects during 1975-2018 can be described

as more effective melting on the western slopes taking place during the afternoon due to combination of more incident solar radiation and warmer air temperature (Evans, 2006). Besides, the north-facing glaciers (including northwest, north and northeast) in this region are more susceptible to area loss ($7.1 \pm 0.7$ km$^2$ or $0.17 \pm 0.02$ km$^2$ a$^{-1}$) than the south-facing glaciers (including southeast, south and southwest), which had a total area recession of $6.9 \pm$ or $0.16 \pm 0.01$ km$^2$ a$^{-1}$. North facing glaciers are mostly clean glaciers that are highly variable with short term changes in temperature and precipitation. Moreover, direct insolation on the northern aspects glaciers lying at the fringe of the Tibetan plateau might have retreated faster under the rising temperature. Here, "*plateau flat surface heating*" effect over the higher elevated semi-arid terrain reported by Brazel and Marcus (1991) can be an effective controlling parameter for the larger amount of glacier area losses from the northern aspects. This is because of the glaciers located on the higher elevations above 5300 m.a.s.l on the north face of Kanchengyao–Pauhunri massif (i.e. Gurudongmar and Tista Khangse) is an extension of Trans-Himalaya of Tibetan plateau, an undulating flat surfaces mostly devoid of vegetation as compared to the southern part in the Thangu valley. Semi-arid higher elevated plateau type topography receives higher insolation and as a result of warmer land surface temperatures (Brazel and Marcus, 1991).

Our field measurements during 1-4 December 2018 (10 am to 1 pm) confirms that mean infrared temperatures of rock (14°C), water (2.7 °C) and grass (11.5°C) on the flat surfaces in the proximity of Gurudongmar lake region (>5150 m) are higher due to more incoming solar radiation than the sloping mountainous terrain near Thangu valley (3900 m) in the south. For example, the mean infrared temperature of rocks, water and grasses near Thangu valley was measured as 4.2°C, 5.1°C and 11.3°C respectively. These recorded data reveals the concept of "*plateau flat surface heating*" effect and a direct impact of the incident solar radiation and effects of shadows on the glacier area change on the northern aspects (Schmidt and Nüsser, 2012).

**Comment:** 397_399: "These glacial fluctuations are controlled by the variations of ISM in summer while the midlatitude westerlies dominates in winters, resulting in a clear seasonality of precipitation which enhanced the glacier melting on the north face of Kanchengyao–Pauhunri massif over the past 43 years (Ali et al., 2018; Benn and Owen, 1998)". This is really not connected to the previous sentences, and even cannot be end like this. Frankly, I could not see any topographic control on glacier changes, see north facing and south facing have almost similar rate. And then how temperature of rock, water and grass comes under topographic influences. This section needs proper discussion on the observations.

**Response**: Sentence deleted from the text. The topographic controlling factors have been elaborated and reframed in the manuscript (New Lines 395-397).

**Comment:** L439: refer Bhambri et al., 2017, there the total number of glaciers are 206 with surge behaviour.

**Response:** Suggestions have been incorporated in the text (New Lines 441-443). "In another study, Bhambri et al. (2017) reported that 221 glaciers in the Karakoram region have surge-type behaviour, covering an area of $7734 \pm 271$ km$^2$ (~43% of the total Karakoram glacierized area)."

**Comment:** L451-454: "The rising temperature contributes to glacier shrinkage over the entire Tibetan Plateau (Fujita and Ageta, 2000; Yao et al., 2012) as well as in the different Himalayan regions (Ageta and Higuchi, 1984; Das and Sharma, 2018; Debnath et al., 2019)". For

Himalayan glaciers regional level ice mass change and perturbation in climate read these articles Bhutiyani et al., 2012; Pratap et al., 2016.

**Response:** Suggestion has been incorporated (New Lines 444-446).

**Comment:** L455: "Moreover, the response of entire Himalayan glaciers is quite sensitive to precipitation, directly or indirectly through the albedo feedback mechanism on the short-wave radiation balance (Azam et al., 2018)". Connect this sentence with previous sentence.

**Response:** Shifted to New Lines 460-463: "Moreover, the Himalayan glaciers are quite sensitive to precipitation, directly or indirectly through the albedo feedback mechanism on the short-wave radiation balance (Azam et al., 2018). Thus, an increasing trend ($\uparrow$) in the annual precipitation (0.639 mm a$^{-1}$) in the region confirms the fact of glacier shrinkage to some extent (Treichler et al., 2019)."

**Comment:** L457: Why it is ironically? I saw this word several times.

**Response:** The word ironically has been changed in both the two places.

**Comment:** Table 2: Use Glacierized instead glaciated. by definition. Glaciated; Covered by glacier ice in the past, but not at present (read Cogley et al., 2011). Glacierized; 'Of a region or terrain, containing glaciers or covered by glacier ice today.

**Response:** Thank you for this valuable suggestion. Replaced the word "Glaciated" with "Glacierized" in all places.

**Added References**

[revised manuscript text omitted]

---

## Author Response (AR1)

*Interactive Reviewer's Comment on* **"Glacier Changes in the Chhombo Chhu Watershed of Tista basin between 1975 and 2018, Sikkim Himalaya, India"** *by* **Arindam Chowdhury et al. 2021**

The authors express their gratitude to the editorial team and the reviewers for their insightful feedback and suggestions to upgrade our manuscript. The detailed specific Author's Responses to the Reviewer'ss Comments are given below. The Author's Responses of the Reviewer's Comments and the revisions in the manuscript are given in blue colour which may be considered as track change.

**Reviewer #1: 09 Mar 2021**

In this manuscript, the background of research is well explained, and what is discussed using the obtained data seems to be appropriate in most cases. But I also think the current manuscript contains various problems, and major revision is needed.

**Reviewer's Comment:** The obtained data about glacier distribution look meaningful in that they are better than the crude data available before. However, it is unclear whether the quality of new data is high enough. More detailed info about data quality should be given - current descriptions in Sections 3.2 and 3.4 are insufficient. For example, orthorectified Sentinel SA images were used as the reference for geometric correction of other satellite images, but no info is given about the error of the orthorectification of the Sentinel images. Whether the presented data are accurate enough is crucial for publication in the journal Earth System Science Data.

**Author's Response:** The data used here, Sentinel 2A, has a much better spatial resolution than any other freely available data sources (i.e., Landsat series) for glacier analysis (Paul et al., 2020). Suggestion has been corrected and incorporated accordingly (P4-L141 and L146-147) as per the analysis done.

**Reviewer's Comment:** A strange thing the authors did is computation of topographic parameters for various years (1975, 1988, 2000, 2010, and 2018) using the SRTM DEM which reflects the topography only in 2000. Therefore, some of computed parameters such as slope do not look meaningful. Please present and use only meaningful data such as the outline and area of each glacier.

**Author's Response:** Thank you for pointing it out. The suggestion has been incorporated accordingly and only spatial changes have been presented using the different multispectral and panchromatic images over the timespan (P5-L175-176).

**Reviewer's Comment:** The quality of English writing is low, with numerous grammatical mistakes, awkward expressions, typing problems etc., although the meaning of sentences is mostly understandable. Three examples of such problems in the earlier half of the abstract are:

L11: "The CCW consists of 74 glaciers" – CCW is a watershed, so "consists of" should be "contains".

L12: The sentence "The change of such glacier outlines obtained ... (2018)" does not have a verb. It is necessary to change "obtained" into "was obtained".

L13: "and" or "but" should be inserted before "by 2018".

Lots of similar problems with English writing can be found throughout the manuscript.

**Author's Response:** English grammar and composition have been revised throughout the text.

**Reviewer's Comment:** Most figures look OK, but they also contain some problems. For example, use of subfigure labels "a1", "a2", and "b1" in Fig. 3 is unusual and strange. Also in its caption, explanation of labels such as IF and CI is written in a random order – please sort them alphabetically.

**Author's Response:** For Fig. 3, suggestions have been incorporated accordingly. The revised Fig. 3 has been added in the manuscript (P28). Fig. 1 has also been revised according to the suggestions.

**Added References**

Paul, F., Rastner, P., Azzoni, R. S., Diolaiuti, G., Fugazza, D., Bris, R. Le, Nemec, J., Rabatel, A., Ramusovic, M., Schwaizer, G. and Smiraglia, C.: Glacier shrinkage in the Alps continues unabated as revealed by a new glacier inventory from Sentinel-2, Earth Syst. Sci. Data, 12(3), 1805–1821, doi:10.5194/essd-12-1805-2020, 2020.

**Revised Figures:**

Some minor changes in the figures and captions as per previous suggestions.

[Figure]

Figure 1. (a) Location of the Chhombo Chhu watershed in the Eastern Himalaya. The watershed boundary is marked in red in the inset within Sikkim, (b) Glacier and glacial lake distribution in the CCW in the Sikkim Himalaya. Inset in black boxes represent the field measurement sites in 2017–2018. The base map used here is SRTM DEM.

[Figure]

Figure 3. Field photographs (2017-18) showing different glaciers and associated geomorphological in the study region (see Fig. 1 for location). (a) Panoramic view of glaciers in the Gurudongmar region; (b-c) Closer view of Gurudongmar and Kangchengyao 2 glaciers; (d) Different morphological types of glaciers in the Tista Khangse region; (e) Closer view of some niche glaciers near Tista Khangse; (f) Tso Lhamo region; (g) Unknown Palaeo Cirque in the Lashar Valley, Thangu. Note: CI, clean ice; GT, glacial tarn; IF, icefall; MCF, Mountain Cliff Face; NLM, new lateral moraine; OLM, old lateral moraine; PDCI, partially debris-covered ice; PGL, proglacial lakes; TK, Tista Khangse; UCLM, unconsolidated Lateral Moraine. The red oval inset surrounding the person represents the scale of the image. (All Photo courtesy: Chowdhury, A. 2017-18).

**Reviewer #2: 05 April 2021**

Review on the manuscript entitle "Glacier Changes in the Chhombo Chhu Watershed of Tista basin between 1975 and 2018, Sikkim Himalaya, India".

**Reviewer's Comments to the authors**

After several read, I find this work very informative, especially the illustration (figures) and data analysed are impressive. This shows the degree of data processing and visibility. One way it may attracts many readers to follow, on the other hand the study has focused only a small catchment containing only 74 glaciers (2018). Considering, the V order basin of the Himalaya, there are >100 of V order basin and 10 times of that the glacierized catchment. Since, remote sensing techniques are well known for its spatial coverage. I would suppose to have this study at least to a basin level with the same efforts.

Thanks for your valuable suggestions.

**General Reviewer's Comments**

Authors could be clearer about what readers can learn from the data about the temporal changes in the length of the glaciers. The following are Reviewer's Comments elaborate on this point.

**Author's Response:** The present study deals only with the glaciers area changes in the watershed. Much of the loss in glacier cover in the frontal sections were observed, therefore the length can be deciphered from the data provided in the Zenodo portal: http://doi.org/10.5281/zenodo.4457183 (Chowdhury et al., 2021).

**Reviewer's Comment:** Basically, this work has focused on the inventory of the glaciers in the Chhombo Chhu Watershed and tried to explain the enhanced retreat rate during recent with metrological data. Climate variability and its impact on glacier changes show no convincing agreement in terms of temporal variations. This must be worth discussing.

**Author's Response:** This sentence has been incorporated in the text (Page 14, Line 482-485) "Since the glacier Author's Response time is of many decades and centuries, the contemporary data may not have any perceptible reflections immediately but over the decades, it is also possible that the present trend of loss in the glaciers mass may be a Author's Response to climatic conditions that existed a few centuries ago, given the mass transfer with respect to velocity, ranging from ~5-25 m a$^{-1}$ in most the Himalayan glaciers."

**Reviewer's Comment:** Similarly: In the last sentence of the abstract, you state that this inventory will provide valuable information for planning the water resources in the Sikkim state of Eastern Himalaya. But such information of temporal changes, planning and execution is not discussed in the text.

**Author's Response:** This sentence is deleted from the text because really it does not have any relation with the present work.

**Reviewer's Comment:** Another thing that was lacking in the manuscript is proper terminology and the literature review. For instance, (PDC) (MDC), mountain (SB) valley (CB) use of such abbreviation is hard to follow as it is newly added and several times used in the manuscript. Use, debris-covered, fully debris-covered, partially debris-covered, mountain glacier, valley glaciers or else. Length change, retreat rate, recessional rate, glacier area loss or glacierized area loss. Fix these words and follow consistently throughout the manuscript.

**Author's Response:** Thank you very much for this valuable suggestion. All of those abbreviations have been replaced with full terminologies throughout the manuscript.

**Detailed Reviewer's Comments**

**Reviewer's Comment:** Abstract: Debris free: use clean ice or clean glacier instead debris free.

**Author's Response:** It is replaced with clean glacier throughout the manuscript.

**Reviewer's Comment:** In fact, it should be mentioned in the method that how you categorized the Clean 'C' type and debris-covered 'D' type glaciers. Partially debris-covered: being quantitative mentioned the % for partial and maximum debris-cover.

**Author's Response**: New Lines 159-1160: It is incorporated in the method section. "We have considered the glacier with more than 50% debris-cover as termed as maximum debris-covered, while less than 50% is called partially debris-covered for a qualitative differentiation and debris-free glacier has been termed as clean glacier."

**Reviewer's Comment:** L18-22: Use quantitative results in the abstract, rather to provide general statement.

**Author's Response:** New Line 19: Suggestions have been incorporated in the abstract.

**Reviewer's Comment:** L29: This sentence is not understandable, or required proper validation such as referencing.

**Author's Response:** New Line 22: This sentence has been reframed.

**Reviewer's Comment:** L30: How and what is the Author's Response indicators, is it Author's Response time of the glacier? If so then try to explain Author's Response indication glaciers in prior sentence.

**Author's Response:** New Line 26: This sentence is added before that sentence to make it more clear: 'Author's Response of negative mass balance is shown in the frontal area as recession supported by identifiable landforms'.

**Reviewer's Comment:** L31: Such comparison in the sentence actually raising curiosity in readers that how large is the difference, is it two-fold or comparably similar specific loss.

**Author's Response:** New Lines 29-32: Suggestions have been incorporated in the text. "Most of the Himalayan glaciers are losing mass at different rates, such as Khatling glacier in the

Garhwal Himalayas and Zemu glacier in the Sikkim Himalayas are retreating at the rate of 88 $\pm$ 0.3 m a$^{-1}$ and 9.1 m a$^{-1}$ respectively (Raj et al., 2017; Rashid and Majeed, 2020)".

**Reviewer's Comment:** L33-34: referencing is more than what was mentioned in the text.

**Author's Response**: New Line 32: Keeping only the first references of 2011 rest of the references are deleted.

**Reviewer's Comment:** L42: Delete 'However'.

**Author's Response:** Deleted and changed the structure of the sentence.

**Reviewer's Comment:** L44: This sentence should come first.

**Author's Response**: New Lines 40-41: The second sentence is shifted at the beginning of the paragraph, as suggested.

**Reviewer's Comment:** L45: Any of the bigger statement need proper citation.

**Author's Response:** New Line 43-45: These sentences have been reframed and suggestion have been incorporated.

**Reviewer's Comment:** L46: sediment related concern needs proper background, why this brings in context to the present study.

**Author's Response:** This sentence is deleted. Thank you very much.

**Reviewer's Comment:** L52-53: It does not make any sense, even cited article not intended to say this statement.

**Author's Response:** This sentence is deleted

**Reviewer's Comment:** L54-56: If this is the case for present study, then all basins are somehow related to the same problems. Here need to state the fundamental of present study by answering how and why.

**Author's Response:** Suggestions have been incorporated, New Lines 49-51 : "The Tista being a trans-boundary river, sustaining a sizable population both in and at the distal reaches of the mountain (Rudra, 2018; Rahaman and Mamun, 2020), our change analysis of the cryospheric parameter would help in long-term planning along the river course, given the current trend."

**Reviewer's Comment:** L58: Delete the whole line. Avoid to use local-level, try to use small scale, regional level, basin level, catchment level etc.

**Author's Response:** This sentence is deleted

**Reviewer's Comment:** L73: Delete, on glaciers since historical past.

**Author's Response:** Deleted this part from the text (New Line 65).

**Reviewer's Comment:** L86-88: Here describe about the climate of the area, not that ISM plays role in glacier fluctuation. Such data article actually needs a greater detail of climate of the study area.

**Author's Response**: New Lines 78-82: Suggestion has been incorporated and the statement has been rewritten by describing the climate of the entire region with additional citation (Murari et al. 2014; Kumar et al., 2020). Moreover, New Lines 84-87 have also been revised.

**Reviewer's Comment:** Suggestion on referencing: the two article Debnath, 2018 and 2019, which were cited 18 times in the manuscript, brings special attention to read both of them, while there were different objectives described, here it was cited for different context. Authors, need to be careful with statements and article citation.

**Author's Response:** Citations of these two articles have been reduced from 18 to 4. Most of the irrelevant sentences related to these references have also been deleted.

**Reviewer's Comment:** L170: please acknowledge the university or institute having ArcGIS 10.2.2 license.

**Author's Response:** We have ArcGIS 10.1 software license at Jawaharlal Nehru University, New Delhi and it is incorporated in the text in acknowledgement. Thanks for pointing it out.

**Reviewer's Comment:** Section 3.6. L201: Provide the aspects of extensive field measurements, means the instrument used or it is visual interpretation and shown with figure 3.

**Author's Response:** The details on the instrument used and reason behind the application has been discussed in details about the nature of filed activities done during the field survey (New Lines 195-197 and 203-206).

"Land surface temperature on the northern and southern part of the watershed for bare rocks, grasses, moraine, water etc. has been measured using the Fluke infrared thermometer. The Digi-Schmidt hammer 2000 has been used to measure the relative hardness of boulders that infers about the relative weathering and relative time of glacier recession from the places."

**Reviewer's Comment:** L206: delete 'dangerous'

**Author's Response:** Deleted

**Reviewer's Comment:** L208: If it is benchmark glacial lakes, there must be name of it.

**Author's Response:** Suggestions have been incorporated in the text (New Line 203).

**Reviewer's Comment:** L212: For minimum size threshold, as per RGI they consider 0.01 km$^2$ the minimum size, but recent inventories have considered 0.02 km$^2$ (Frey et al., 2012; Chand and Sharma, 2015). This information should come in method section with reference.

**Author's Response:** Included this sentence in the last part of the first paragraph under point 3.3 (glacial inventory mapping) (New Lines 154-155)

**Reviewer's Comment:** L220-225: This will be better while comparing with other studies in discussion section, so shift it.

**Author's Response:** Shifted in the point 5.3 (Regional Comparison with other Himalayan basins)

**Reviewer's Comment:** L286-289: This can be shift to discussion section. And delete Fig. 10c showing the retreat map of the study area (1975–2018).

**Author's Response:** Shifted in the point 5.1 last sentence (Comparative evaluation of glacier inventory in the CCW- New Lines 350-351). The caption of Fig. 10c is revised and cited in New Line 275.

**Reviewer's Comment:** L290: Describe the results and then followed by the referred figure.

**Author's Response:** Figure number has been shifted after results (New Line 282)

**Reviewer's Comment:** General: L300: Better term is clean glacier or 'C' type, not debris free DF, and please used full word for terminology having definitions. It is interesting that having enhanced retreat rate of the glaciers in this catchment, still have higher number of clean glaciers. Ok what is the criteria set for clean glaciers in this study? PDC: same here, how much debris cover (in % either for the total glacier area or for total ablation area) was categorized for partial debris-covered glaciers.

**Author's Response:** Suggestions have been incorporated and the sentence is revised according to the suggestion (P9-L314). All of those abbreviations have been replaced with full terminologies throughout the manuscript. The criteria are based on qualitative differentiation which is already mentioned in the method section (New Lines 159-160)

**Reviewer's Comment:** L305: "mainly due to lower terminus elevation" This is not the reason for recession, it is a mechanism that glaciers having longer length and lower terminus position has receded at higher rate due to higher negative mass balance at the lower elevation.

**Author's Response:** Deleted "mainly due to lower terminus elevation" and added one sentence 'Due to the glaciers having a longer length and lower terminus position, it has receded at a higher rate due to higher negative mass balance at the lower elevation." (New Lines 296-297)

**Reviewer's Comment:** L313-314: This is the process observed but the deferential processes is something else, which I described in the above Reviewer's Comment. So rather, you may show the observation instead the reasons.

**Author's Response:** The sentence is changed like this "It is found that the maximum glacier area loss is taking place at the lower terminus elevation and comparatively lesser slope of large valley (simple basin) glacier than the other morphological types." (New Lines 304-305)

**Reviewer's Comment:** 5.1 CCW: Chhombo Chhu Watershed

**Author's Response:** Corrected it (New Line 325)

**Reviewer's Comment:** 335-345: For this comparison between inventories of glaciers, providing detail only in Table will not justify. For valid comparison and stating the discrepancies of other inventory you might have to provide a figure with glacier outline. I think it is available for all datasets.

**Author's Response:** Suggestions have been incorporated (New Line 338)

Figure 13. Glacier inventory: (a) overlay of glacier outlines of different inventory on Landsat 7 ETM+ 8 November

[Figure]

2000; (b) Slope map (in °) showing the misinterpretation of glacier boundary and ice/sub-watershed divides. Dotted black lines represent the watershed divide.

**Reviewer's Comment:** 348-353: First, Authors have selected wrong dataset for GSI inventory, so check with Sangewar and Shukla, 2009. Second, that was the era, when Geological Survey of India has make larger efforts to provide a detail study of Indian Himalayan glaciers. So, I suggest to follow their glaciological terminology and the best out of the old literature. Avoid to say misinterpretation, rather you can provide uncertainties.

**Author's Response:** New Line 329: Error has been corrected as we have now considered the published Glacier Atlas of India by Raina and Srivastava (2008) only. Suggestions (such as full terminologies for morphological types) are well taken and incorporated accordingly throughout the text.

**Reviewer's Comment:** 384: December: provide years and number of days with time of measurement. The surface temperature has diurnal variability. So comparing morning and evening may lead to misinterpretation.

**Author's Response:** Suggestions have been incorporated (New Line 384).

**Reviewer's Comment:** L394-397: Though, the difference is very much close, still authors need to discuss the reason in here, rather to provide results.

**Author's Response**: Suggestions have been incorporated. The probable controlling factors have been identified and elaborated in the manuscript (New Lines 377-397).

The difference of glacier area loss on the western ($2.8 \pm 0.2$ km$^2$ or $0.06 \pm 0.005$ km$^2$ a$^{-1}$)

and eastern ($1.1 \pm 0.1$ km$^2$ or $0.06 \pm 0.003$ km$^2$ a$^{-1}$) aspects during 1975-2018 can be described as more effective melting on the western slopes taking place during the afternoon due to combination of more incident solar radiation and warmer air temperature (Evans, 2006). Besides, the north-facing glaciers (including northwest, north and northeast) in this region are more susceptible to area loss ($7.1 \pm 0.7$ km$^2$ or $0.17 \pm 0.02$ km$^2$ a$^{-1}$) than the south-facing glaciers (including southeast, south and southwest), which had a total area recession of $6.9 \pm$ or $0.16 \pm 0.01$ km$^2$ a$^{-1}$. North facing glaciers are mostly clean glaciers that are highly variable with short term changes in temperature and precipitation. Moreover, direct insolation on the northern aspects glaciers lying at the fringe of the Tibetan plateau might have retreated faster under the rising temperature. Here, "*plateau flat surface heating*" effect over the higher elevated semi-arid terrain reported by Brazel and Marcus (1991) can be an effective controlling parameter for the larger amount of glacier area losses from the northern aspects. This is because of the glaciers located on the higher elevations above 5300 m.a.s.l on the north face of Kanchengyao–Pauhunri massif (i.e. Gurudongmar and Tista Khangse) is an extension of Trans-Himalaya of Tibetan plateau, an undulating flat surfaces mostly devoid of vegetation as compared to the southern part in the Thangu valley. Semi-arid higher elevated plateau type topography receives higher insolation and as a result of warmer land surface temperatures (Brazel and Marcus, 1991).

Our field measurements during 1-4 December 2018 (10 am to 1 pm) confirms that mean infrared temperatures of rock (14°C), water (2.7 °C) and grass (11.5°C) on the flat surfaces in the proximity of Gurudongmar lake region (>5150 m) are higher due to more incoming solar radiation than the sloping mountainous terrain near Thangu valley (3900 m) in the south. For example, the mean infrared temperature of rocks, water and grasses near Thangu valley was measured as 4.2°C, 5.1°C and 11.3°C respectively. These recorded data reveals the concept of "*plateau flat surface heating*" effect and a direct impact of the incident solar radiation and effects of shadows on the glacier area change on the northern aspects (Schmidt and Nüsser, 2012).

**Reviewer's Comment:** 397_399: "These glacial fluctuations are controlled by the variations of ISM in summer while the midlatitude westerlies dominates in winters, resulting in a clear seasonality of precipitation which enhanced the glacier melting on the north face of Kanchengyao–Pauhunri massif over the past 43 years (Ali et al., 2018; Benn and Owen, 1998)". This is really not connected to the previous sentences, and even cannot be end like this. Frankly, I could not see any topographic control on glacier changes, see north facing and south facing have almost similar rate. And then how temperature of rock, water and grass comes under topographic influences. This section needs proper discussion on the observations.

**Author's Response**: Sentence deleted from the text. The topographic controlling factors have been elaborated and reframed in the manuscript (New Lines 395-397).

**Reviewer's Comment:** L439: refer Bhambri et al., 2017, there the total number of glaciers are 206 with surge behaviour.

**Author's Response:** Suggestions have been incorporated in the text (New Lines 441-443). "In another study, Bhambri et al. (2017) reported that 221 glaciers in the Karakoram region have surge-type behaviour, covering an area of 7734 ± 271 km$^2$ (~43% of the total Karakoram glacierized area)."

**Reviewer's Comment:** L451-454: "The rising temperature contributes to glacier shrinkage over the entire Tibetan Plateau (Fujita and Ageta, 2000; Yao et al., 2012) as well as in the different Himalayan regions (Ageta and Higuchi, 1984; Das and Sharma, 2018; Debnath et al., 2019)". For Himalayan glaciers regional level ice mass change and perturbation in climate read these articles Bhutiyani et al., 2012; Pratap et al., 2016.

**Author's Response:** Suggestion has been incorporated (New Lines 444-446).

**Reviewer's Comment:** L455: "Moreover, the Author's Response of entire Himalayan glaciers is quite sensitive to precipitation, directly or indirectly through the albedo feedback mechanism on the short-wave radiation balance (Azam et al., 2018)". Connect this sentence with previous sentence.

**Author's Response:** Shifted to New Lines 460-463: "Moreover, the Himalayan glaciers are quite sensitive to precipitation, directly or indirectly through the albedo feedback mechanism on the short-wave radiation balance (Azam et al., 2018). Thus, an increasing trend (↑) in the annual precipitation (0.639 mm a$^{-1}$) in the region confirms the fact of glacier shrinkage to some extent (Treichler et al., 2019)."

**Reviewer's Comment:** L457: Why it is ironically? I saw this word several times.

**Author's Response:** The word ironically has been changed in both the two places.

**Reviewer's Comment:** Table 2: Use Glacierized instead glaciated. by definition. Glaciated; Covered by glacier ice in the past, but not at present (read Cogley et al., 2011). Glacierized; 'Of a region or terrain, containing glaciers or covered by glacier ice today.

**Author's Response:** Thank you for this valuable suggestion. Replaced the word "Glaciated" with "Glacierized" in all places.

**Added References**

[revised manuscript text omitted]

---

## Author Response (AR2)

**Comments to the Author:**

I edited the abstract and text a bit. I do not recommend the use of has been and have been as they are past continuous. You have already completed the study.

**Authors' Response:**

Thank you very much for your positive comments. We have replaced 'has been' and 'have been' from the whole manuscript with 'is' and 'are', as suggested.

In the abstract part we have changed 'Fully' with 'Maximum', because throughout text we have used 'Maximum Debris Covered Glaciers' (MDC).

**Non-Public Comments to the Author:**

The paper needs careful editing before publication. The paper documents changes taking in the Himalayas. It should be a good resource for future studies.

**Authors' Response:**

Thank you very much for your comments. We have rechecked and revised the whole manuscript.